# Representing and Experiencing Islamic Domes: Images, Cosmology, and Circumambulation

**Shunhua Jin**

School of Philosophy, Fudan University, Shanghai 200433, China; shjin18@fudan.edu.cn

**Abstract:** Lindsay Jones developed the concept of "ritual-architectural event", according to which the meaning of a sacred building depends upon the participant's experience of it in the course of the rituals they perform. Starting from such approach, and taking the Islamic dome as my subject-matter, I examine the correlations that link architectural forms, ritual performance, and participants' experience into a whole. I first survey a corpus of images related to domes in two types of manuscripts (poetry, and pilgrimage narratives), showing how these images suggest cosmological patterns. The second part unfolds these representations, proceeding from cosmology to ritual. The third and last part focuses on circumambulation as the ritual experience that best embodies the previously identified cosmological patterns. The connection between the three dimensions discussed here is ascertained by the fact that the combination of circle and square structures relates both to Islamic graphic representations and ritual practices. An aesthetic/spiritual experience is awakened both in the mind and in the bodily senses of the viewer/practitioner: When Muslims stand under a dome, in front of the mihrab, thus facing Mecca, and when they behold the dome under which they stand, the view of this circular space possibly translates into a kind of mental and spiritual circumambulation. The conclusion suggests that the meaning attached to sacred architecture places is triggered by a complex of interactions between patterns referred respectively to the mind, bodily actions, and cultural settings.

**Keywords:** architectural experience; circumambulation; Islamic cosmology; Islamic dome; pilgrimage; sacred architecture

## 1. Introduction and Theoretical Framework

Sacred architecture obeys ritual stipulations. For instance, the qibla wall in mosques must be oriented towards Mecca, thus asserting the direction towards which to pray. Reciprocal rituals regulate believers' behavior in sacred places. In the last two decades or so, the various ways ritual and architecture have been correlated has drawn the attention of many scholars (Miller 2007; Wescoat and Ousterhout 2014; Ragavan 2013; Jones 1993a, 1993b, 2000a, 2000b). Lindsay Jones, one of the earliest theorists on ritual and architecture, has propounded the concept of the "ritual-architectural event" in order to explore the meanings attached to religious buildings (Jones 2000a, 2000b). From the perspective Jones developed, the meaning of a building depends upon the situation of the building, with the participant and rituals involved. In other words, as participants perform rituals within the building, they actualize the meaning attached to the latter. In the word of Jones, the meaning of the building must always be a meaning for someone, at some specific time, and in some particular place (Jones 2000a, p. 41).

Based on Gadamer's hermeneutics of art, Jones' perspective not only brings participants and rituals into our understanding of what sacred architecture is about, but also directs our attention to the flux into which the meaning of a building is necessarily inserted: the meaning of a building "not occasionally, but always" surpasses the intentions that presided its construction (Jones 1993a, p. 212). However, Jones' promising hermeneutical approach, which emphasizes "the human experience of buildings", also brings out new challenges, among which the following ones are of particular concern for this article.

How are we going to distinguish, as Jones wants us to do, between the "familiarity" induced by the rituals ordained by the building and the "unfamiliarity" whose surge is allowed by the latent presence of an "otherness" in the same building? (See notably Jones 1993a, 1993b). Is it possible that the unfamiliar be felt without the participants reaching full consciousness of an underlying "otherness"? On the one hand, Jones describes a ritual-architectural event as necessarily including (1) the actual built form; (2) human beings bringing in expectations, traditions, and religious opinions; and (3) the ceremonial occasion that brings buildings and people into to-and-fro involvement with one another. On the other hand, Jones uses the concept of "play" (borrowed from Gadamer), a play that relates people with buildings in order to underline the dynamism and flexibility of the response emanating from the people experiencing the building (Jones 1993a, pp. 214–15). Jones' mentions of the built-in form and of human expectations clearly show that the meaning of a building is inserted into a given cultural context. The last point is the ceremonial occasion, the ritual introduces an additional dimension: Following Gadamer and Jones, one may describe the ritual occasion as providing participants with "the rules of the game" rules that could not master the mere visitor of a given sacred building if (s)he is an outsider to the "game" played within it. Religious rituals are normally performed according to a set of regulations. For example, Muslims should face the Kaʿba to worship. Meanwhile, as Jones points out, religious buildings are built upon conventional forms that may predate these ritual stipulations, and architecture has its own constraints, which may trigger specific experiences. I want to argue in this paper that when a viewer/participant inhabits a building on a ceremonial occasion, the unfamiliar experience rises partly from the fact that familiar forms are mobilized for ritual purposes loaded with "rules". And the "rules" speak to the subconscious of the viewer/participant, awaken in her/him an "otherness", and is conducive to a flux of meaning that somehow "overwhelms" the participant.

The question is made more acute by the fact that ritual is an essential expression of religious and cultural structures. According to Catherine Bell: "Ritual is a cultural and historical construction that has been heavily used to help differentiate various styles and degrees of religiosity, rationality and cultural phenomenon" (Bell 2009, p. ix). Ritual patterns are confirmed and reinforced by the fact that religious buildings have a relatively stable form: mosques, although they do not have a fixed format, do have conventional features. In a sacred space, believers receive perceptions, messages, and imagery anchored into a cultural system. This is why I focus on the significance of the participants' engagement with architecture in "a continuity of tradition", as Gadamer and Jones would have it. Both rituals and architecture obey rigorous patterns, even if, once in a while, they take some distance from their observance. At the same time, and while I agree that the meaning of a building goes beyond the structures provided by the builder, and that "the surplus of meaning" is produced in dialogue and play between humans and buildings, I suggest that the same "surplus of meaning" has actually to do with the inscription of situational behavior into pre-existing patterns. "Novelty" or "otherness" in some cases, are connected to the way the "tradition" is enacted and experienced, through ritual observances.

In other words, while working in the framework provided by the concept of "ritual-architectural event", I qualify the latter by stating the following points: (1) the participant to a ritual event does not let her/his imagination float freely: ritual patterns both trigger and direct the interpretation given to the space-time in which (s)he is inserted. (2) This leads us to draw a sharp boundary between a "participant" and an "onlooker" to a "ritual-architectural event". (3) At the same time, the feeling of "otherness" is not experienced only by onlookers: putting into motion hidden, collective patterns, the ritual structure awakens in the participant's associations that complexify the sense of tradition and continuity.

Taking the Islamic dome as our subject-matter, I will examine the correlation that links architectural forms, ritual performance, and participants' experiences into a whole. Domes emerged early in Islamic architecture and are regularly construed as an element in front of the mihrab in a mosque. The earliest remaining Islamic monument, The Dome of the Rock in Jerusalem, also the first domed building in Islamic architecture, was completed in 691 A.D.

(Ettinghausen et al. 2001, pp. 15–20). And the dome set in front of the mihrab in mosques appeared in the 9th century. For example, the dome of the great mosque of Kairouan was made in 862 (Bayati 1985). In the 11th century Seljuk Iran, the introduction of a maqṣūra (enclosure) in front of the mihrab of the hypostyle mosque helped to transform the skyline of Persian towns, characteristically punctuated with domes (O'Kane 2020; also see Creswell 1914, 1915; Hillenbrand 1976). In the Ottoman empire, after the imperial architect Mimar Sinan created major domed mosques, the dome became a common architectural form in the Ottoman territory (Neçipoğlu 2005). Two 16th century Ottoman scholars, Sai Mustafa Çelebi (d. 1595) and Mustafa ibn Celal, state that the Süleymaniye's four minarets and its dome represent the Prophet who is the "Dome of Islam" (Neçipoğlu-Kafadar 1985, p. 106). The large territory of the Ottoman empire spread the Ottoman domed architecture to North Africa, Europe, and Arabian Peninsula. This introduction cannot cover the complex development of domes throughout the different regions of the Islamic world, but the usage of the dome in Islamic architecture (including in the mosque, palace, mausoleum, madrasa, and so on) is remarkable. The dome is loaded with significance in Islamic societies. If we ask a child to draw a mosque nowadays, even though a dome is not a religiously required element of this sacred building, (s)he will probably draw a dome on the top of it.

Representations of the dome appear abundantly in Islamic manuscripts, and these representations will mediate and direct the inquiry conducted here. The reason for which this article will study the images of the dome rather than proper buildings can be stated as follows: as elucidated by the "ritual-architectural event" approach, the meaning of the building depends upon the participants, and it takes shape in the way they experience the building. Jones' suggestion of a hermeneutical dialogue/play involving building, human, and ritual brings the possibility to associate or evoke architectural form in other patterns of images, which leads to shifts in meaning and personal transformation. In our case, the Muslim world offers an abundance of manuscripts that contain images holding stable symbolic and cultural patterns. Within this comprehensive iconographic system, representations of the dome reveal the religious and cultural associations that may have surged in the mind of the participants during the course of sightseeing and worshipping.

The material discussed below is taken from medieval Persian and early modern Ottoman manuscripts, on the one hand, and from contemporary visual art inscribed into the Islamic tradition, on the other. This already points towards the fact that our discussion will suggest that the experience of architecture crisscrosses time and space. I will first survey a corpus of images related to domes in two types of manuscripts (poetry, and pilgrimage narratives), showing how these images suggest cosmological patterns. In the two following parts, I will further unfold and interpret these representations: I will proceed from cosmology to ritual, and I will make circumambulation the ritual experience that best embodies the previously identified cosmological patterns. My conclusion will associate these various elements, suggesting a way to depict structures of meaning associated with the fact of experiencing one's setting into sacred architectural places.

## 2. Image of Domes in Manuscripts

### 2.1. Poetry Manuscripts

The Persian poem *Haft Paykar* (*Seven Portraits*) authored by the 12th-century poet Nizami Ganjavi, develops the imagery of the dome, and the manuscripts of this work unfold such imagery. In the story, seeing the portraits of seven princesses who later become his brides, King Bahrām Gūr, fascinated by them, asks Shīda to build a palace with seven domes for hosting the princesses. Shīda is a great master skilled in painting, sculpture, architecture, calligraphy, and in sciences as well; the poem says: "Physics, geometry, astronomy—all in his hand, was like a ball of wax" (Nizami 1924, p. 110). With all these skills and knowledge, "he made the dome so heavenly that no one could distinguish it from heaven" (Nizami 1924, p. 112). Oleg Grabar inscribes the meaning of the domes in this poem into an "ideology of pleasure" (Grabar 1990, p. 18). Yet, as we will now see from the analysis of some features of this poem, the aesthetic experiences and context in

Nizami's narrative associated with the domes considerably beyond pleasure, and happens to be extremely rich in cosmological symbolism.

Seven princesses come from different parts of the earth, and each receives a dome of a particular color. The seven domes represent the 7 days of the week and are governed by seven planets, which leads to the following associations:

1st. Hindu. Saturn. Black dome. Saturday.

2nd. Chīn (China, Turkestan). Sun. Yellow dome. Sunday.

3rd. Khwārazm. Moon. Green dome. Monday.

4th. Siqlab (Slavonia). Mars. Red dome. Tuesday.

5th. Maghreb (North Africa). Mercury. Turquoise dome. Wednesday.

6th. Rüm (Byzantium). Jupiter. Sandal dome. Thursday.

7th. Persia. Venus. white. Friday.

(Ganvaji 2015, pp. xvi–xix)[1]

Bahrām visits one princess, in the dome assigned to her, each day of the week, and each princess hosts a feast and tells stories to the king. In a 15th-century Persian painting, the king is standing in a room where he sees portraits of the seven princesses. The painter depicted the domes with suggestive colors above each princess' portrait. The room is in a curved shape to enhance the decorative factor of the illustration (Figure 1).

The number seven, constantly used by Nizami in his poetry, has evolved as a symbolic number since ancient Persia and has continued to be loaded with symbolic meanings in the Islamic tradition (Schimmel 1994, pp. 142–50)[2]: Allah created seven heavens in layers (Qurʾan 65:12, 71:15), and the Prophet saw seven heavens during Miʿrāj. In one of the most comprehensive medieval encyclopedias, *the Epistles of the Brethren of Purity* (*Rasāʾil Ikhwān al-Ṣafāʾ*), the number seven is presented as the first perfect number (Nasr 1978, p. 97) and also refers to the seven planets. Moreover, both in pre-Islamic and Quranic sources, nearly all authors of geography treatises accepted that there are seven seas and seven climates in the world (Nasr 1978, pp. 87–88).[3]

Several scholars have focused upon the features of the number seven and its association with the textual structure of *Haft Paykar*. Peter Chelkowski comments: "Combined of three and four, seven is geometrically expressed as a triangle and a square." (Chelkowski 1975, p. 113). Likewise, Julie Meisaimi sees the poem as structured in three parts: the body; the soul; and the union of body and soul. Bahman's spiritual journey, divided into four phases, begins when seeing the portrait of the Seven Princesses in the room, and ends when walking towards the cave, the whole process constructing a circle from the beginning to the end (Meisaimi 1997).

Slightly earlier than Nizami, the 10th-century Persian astronomer and philosopher al-Bīrunī (973–1048), in *The Book of instruction on the elements of the art of astrology*, indicated the colors, tastes, and smells of the seven planets. For instance, "Saturn is extremely cold and dry. The greater malefic. Male. Diurnal. Disagreeable and astringent, offensively acid, stinking. Jet-black, also black mixed with yellow, lead color, pitch-dark"; "Sun is hot and dry, the heat predominant. Maleficent, when near, is beneficent at a distance. Male. Diurnal. Penetrating. Pungent, shining reddish-yellow. Its color is said to be that of the lord of the hour" (Biruni 1934, pp. 396–401).

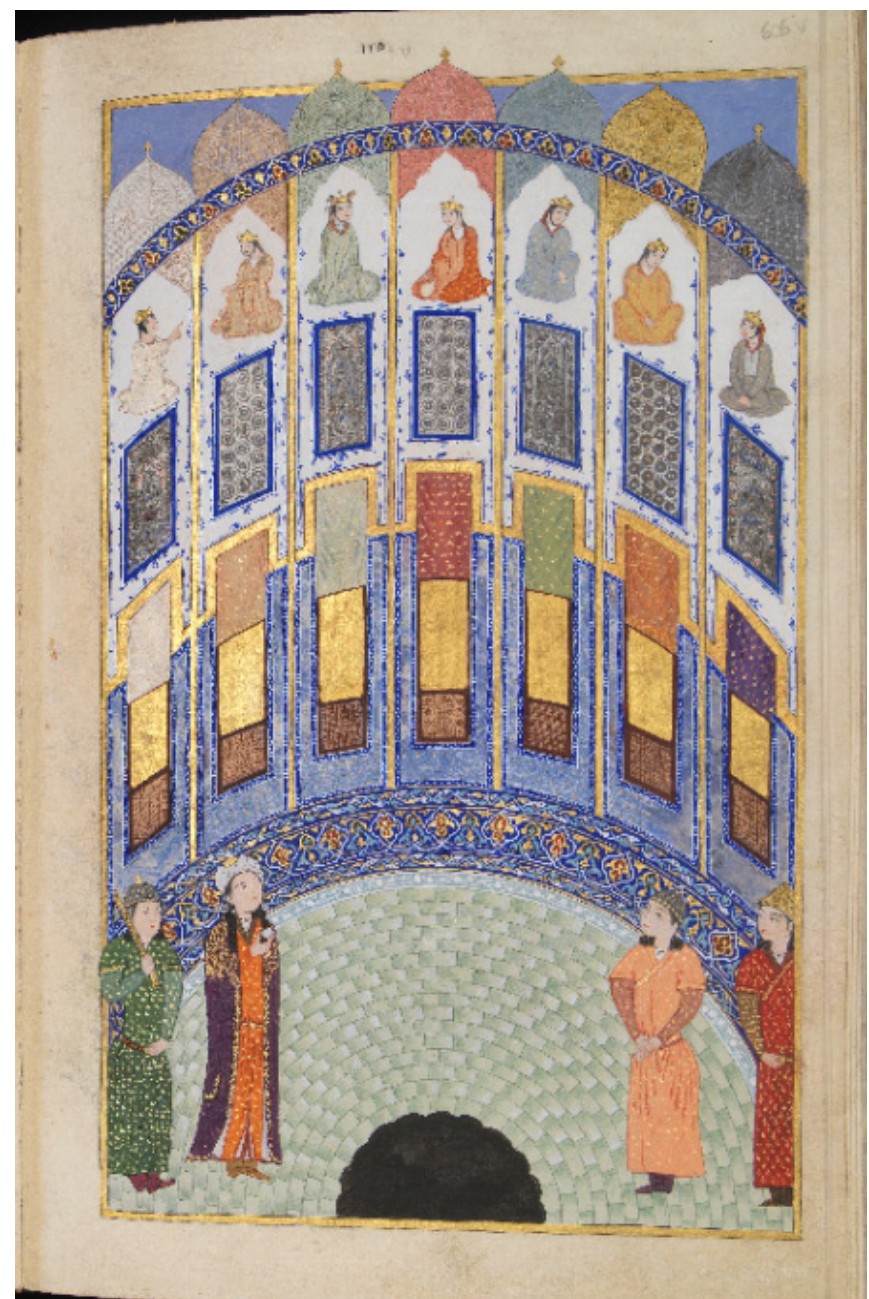

**Figure 1.** Miniature from the Anthology of Sultan Iskandar, Shiraz, Persia, Timurid period, 1410–11. © Calouste Gulbenkian Foundation. 2022.

al-Bīrunī's descriptions of the planets' color are more intricate than the way Nizami's poem associates the color of the domes related to the planets, and the associations extend to soils, buildings, countries, jewels, foods, animals, drugs, etc. Interestingly, al-Bīrunī also associated the planets with architectural types, such as:

> "Saturn: Underground canals and vaults, wells, old buildings, desolate roads, lairs of wild beasts [ . . . ] Jupiter: Royal palaces, mansions of the nobility, mosques, pulpits, Christian churches and synagogues [ . . . ] Sun: Kings' and sultan's palaces [ . . . ]", and so on. (Biruni 1934, pp. 407–8)

Although al-Bīrunī does not single out the dome in the architectural–planet relation, his text hints at the metaphorical manner in which Nezami was associating the seven domed palaces with the planets. At the same time, al-Bīrunī's astronomy book and Nizami's poetry

are two different types of text. Hence, the fact that the planets awaken different suggestions from the one to the other. However, the correlation between architectural forms (the dome), astronomy (the planets), the calendar (the days of the week), and geography (the princesses of different regions) does not accidentally appear in Nizami's text. A consistent system of Islamic cosmology lies behind it. Cross also made a comparison between al-Bīrūnī schematic map of the seven climates and the world of the *Haft paykar* (Cross 2016, p. 60).

Cosmology is the science of the cosmos—its origin, structure, components, order, and governing laws (Akkach 2005, p. 1; Nasr 1978). Cosmology also includes the discussion of ecology and society, of human and nonhuman beings and forces, both perceptible and imperceptible, as they all constitute parts of that universe of commonly shared representation (El-Sayed 2002, p. 2). Nizami consciously made the dome a combination of astrology and geography located in cosmology. Reciporocally, in his poetry, he endows the architect Shīda with the ability to combine the knowledge of cosmology with architecture (Shīda is also skilled in painting, sculpture, and calligraphy).

In classical Islamic thought, arts as the skills mastered by human hands lie within the category of ʿilm (plural form: ʿulūm), a category that refers to science or knowledge, and was first applied to certain religious sciences. It came to designate, generally in the plural (ʿulūm), the new areas of knowledge that were developing in Islamic society as well as many arts and crafts, with almost no specificity (Vílchez 2017, p. 101). Therefore, in poetry, only a master like Shīda whose knowledge (ʿilm) includes cosmology can build the seven palaces. The concept of ʿilm includes epistemological, moral, and socio-religious dimensions (Akkach 2019, p. VI; Rosenthal 2007).[4] Going one step further, in pre-modern Islam, ʿilm is invested with a comprehensive meaning that combines art, science, and religion (Akkach 2019). The references associated to the seven-domed palace in *Haft Pakyar* constitute an additional example of the extension of the concept. Seen from this perspective, the construction of the domes and the experience that the king acquires from them synthesize astronomical, arithmetical, architectural, and decorative knowledge. Remarkably, Nizami adopted the dome as an architectural form so as to synthesize such knowledge with cosmology in his background.

### 2.2. Pilgrimage Texts

In addition to illustrations found in poetry, domes frequently appear in some pilgrimage manuscripts of the Ottoman period, mainly in reference to monumental buildings such as holy tombs and shrines. Two notable pilgrimage and prayer manuscripts have been transcribed for hundreds of years. One of them is *Futūḥ al-Ḥaramayn* (*The Revelation of the Holy City*), and the other is *Dalʾil al-khayrāt* (*The Way to Happiness*) (Roxburgh 2011, pp. 37–38). A 16th-century writer Muḥyī al-Din Lārī (d. 1526 or 1527) completed the writing of *Futūḥ al-Ḥaramayn* in 1506, and he dedicated the book to Muzaffar ibn Mahmud Shah, the ruler of Gujarat in India. It is a Persian-language Hajj guide about the holy places to be visited, with virtual instructions added, usually with illustrations of Mecca and Medina. Although Jerusalem is not included in the original text, the Haram al-Sharif appears in some later copies, making for a total of 19 sites of pilgrimage (Milstein 2006, pp. 167–94; Roxburgh 2011, p. 38). The primary pilgrimage site is al-Masjid al-Haram (the Great Mosque of Mecca), the Kaʿba, which is the centre of Islam.

The image of al-Masjid al-Haram represents a rectangle seen from the top, joining two enclosures with arched gates around, which corresponds to the mosque's wall and corridor. This setting transforms the image into a diagram. However, the illustrator drew the picture according to the layout of the building and its surrounding. As Rachel Milsten pointed out, a copy of *Futūḥ al-Ḥaramayn* made in 1568–69 A.D. (Israel Museum, IM 838.69) depicts the seventh minaret built in 1565–66 by order of Suleiman the Magnificent. Moreover, the arched gate around the rectangle has a flat roof without domes in this illustration, while in another manuscript made in 1573 (L. A. Mayer Museum, no. Ms. 34–69), the roof of the wall holds domes on the top (Milstein 2006, p. 173). Two illustrations of al-Masjid al-Haram were collected in Met (Metropolitan Museum of Art), one that was made in the

mid-16th century without domes on the enclosures (Figure 2a), while the other, copied in the late 16th century, has domes (Figure 2b). The details of the description of the building became the reference for determining the period of the manuscript.[5] Taking into account the difference in details, it can be said that the diagrams made of *Futūḥ al-Ḥaramayn* lie between convention and factuality. The transcription of pilgrimage guides, as well as pilgrimage certificates with Meccan iconography, testifies to the fact that, in later periods, such representations became schematic drawings, which enables us to identify a similar structure among this array of images.

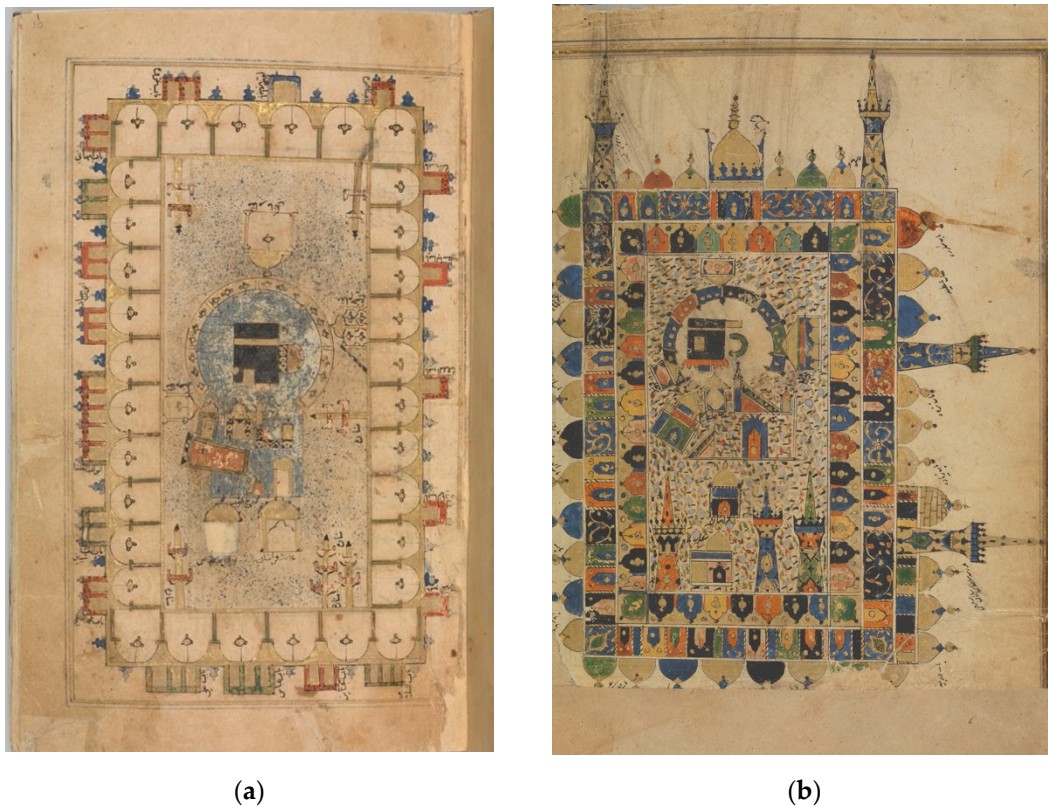

(**a**)                                                          (**b**)

**Figure 2.** (**a**) Illustration of al-Majid al-Haram, *Futūḥ al-Ḥaramayn*, mid-16th century. The Metropolitan Museum of Art, 32.131, 10r. photo in public domain. (**b**) Illustration of al-Majid al-Haram, *Futūḥ al-Ḥaramayn*, late-16th century. The Metropolitan Museum of Art, 2009.343, p. 40. photo in public domain.

These illustrations of *Futūḥ al-Ḥaramayn* contain a series of signifiers: the Kaʿba is a black square, the minbar is a staircase by its side. And the domes seem to refer to monumental buildings. Moreover, the names of the monument being represented are frequently written next to their diagrams. These names make use of diverse words referred to domed buildings (Figure 3b), such as "qubba" (Arabic: قبة), "gonbad" (Persian: گنبد) or "Mashhad" (Persian: مشهد).[6] The birthplaces of Muhammad, Ali, Abu Bakar, Fatima, and other saints are also characterized by domes (Figure 3a); as we discussed above, the images of *Futūḥ al-Ḥaramayn* rely on the factuality of architecture, indicating that in the 16th century, the Islamic holy land was landmarked by domes that had a clear monumental and sacred connotation.

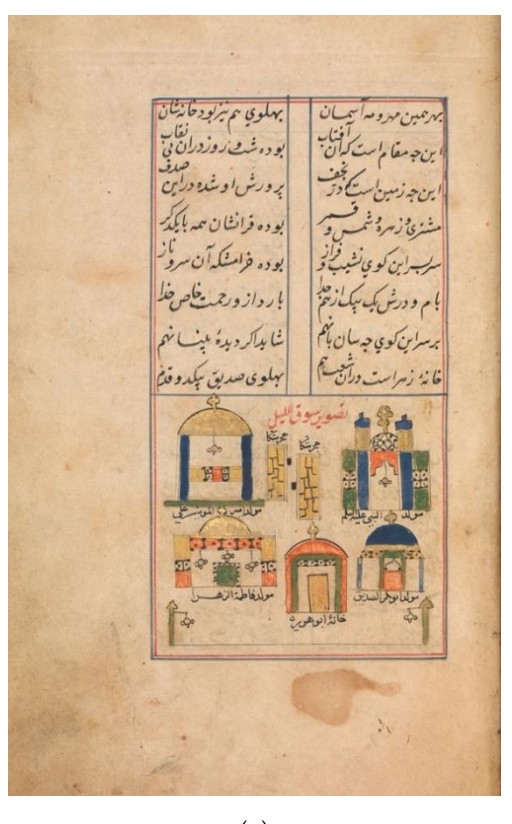 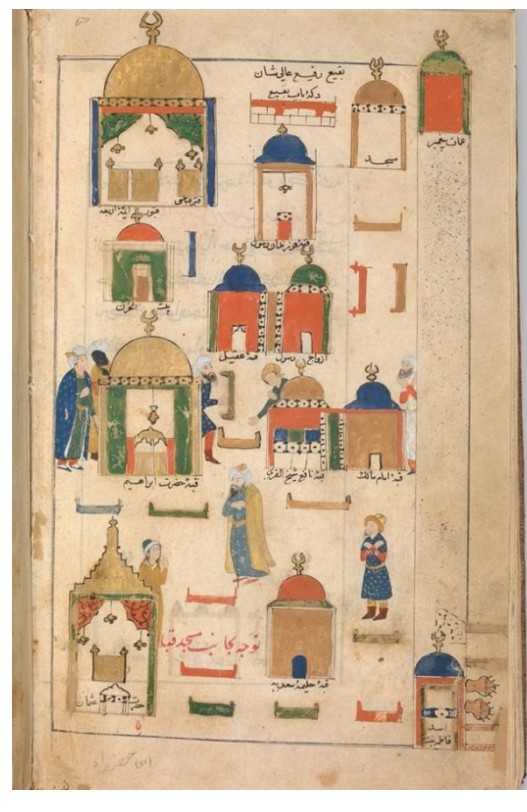

(**a**)                    (**b**)

**Figure 3.** (**a**) Illustration of *Futūḥ al-Ḥaramayn*, The Metropolitan Museum of Art, 32.131, 22r. photo in public domain. (**b**) Illustration of *Futūḥ al-Ḥaramayn*, The Metropolitan Museum of Art, 32.131, 24r. photo in public domain.

In the paintings associated with Nizami's poetry and in the diagrams of religious manuscripts, the domes are represented chiefly in a sideways form with a pointed arch shape. Sometimes these representations even characterize the regional style of the domes. For example, domes are onion-shaped in some pilgrimage manuscripts from India. One exceptional case is an illustration of the Dome of the Rock in Jerusalem made from the top, using a diagram to indicate the holy site, similar to the description of the first and second most holy sites in Islam, Mecca, and Medina in *Futūḥ al-Ḥaramayn*. In this Ottoman manuscript (Figure 4), the illustration page is separated into two parts, the above is the Al-Aqsa Mosque, and below is the Dome of the Rock, depicted as an octagon. Furthermore, another version of this pilgrimage site, the Dome of the Rock, appears like a circle-shaped diagram collected in the University of Michigan Library (Isl. Ms. 397, f. 47r). (Gruber 2014)[7] The rationale behind this unique description of the Dome of the Rock might come from two parts. On one hand, it inherited architectural traditions from the eastern Mediterranean, particularly from local Christian material traditions, for instance, the Church of the Kathisma, a 5th-century Byzantine church located between Jerusalem and Bethlehem (Avner 2010, pp. 31–49; Ettinghausen et al. 2001, p. 17). On the other hand, the diagrammatical representation of the Dome of the Rock in the 16th century pilgrimage manuscript is needed to be octagonal to reflect the unique religious scenario, like Mecca and Medina. It is possible that Jerusalem, as the site of the Last Judgment, is directly associated with the fact of ascending to Heaven, the circle being used so as to emphasize this symbolism.

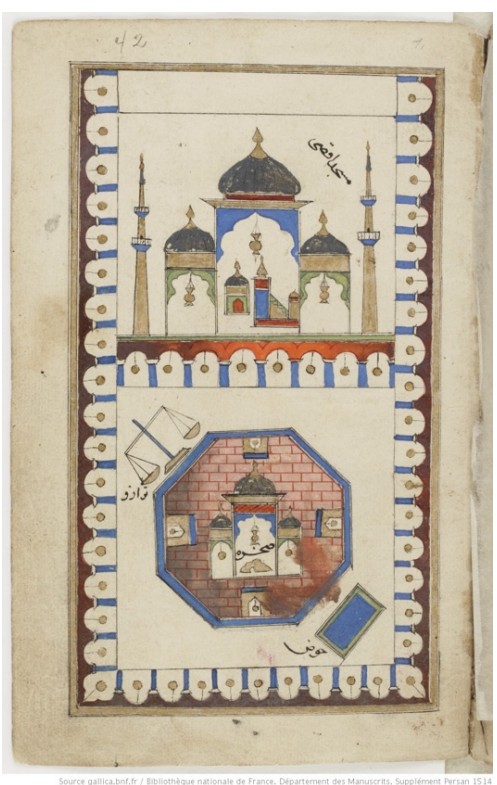

Source gallica.bnf.fr / Bibliothèque nationale de France, Département des Manuscrits, Supplément Persan 1514

**Figure 4.** Jerusalem. Muḥyī al-Dīn Lārī, *Futūḥ al-Ḥaramayn*, 1577 A.D., Bibliothèque nationale de France, Paris, Suppl. Pers. 1514, folio 42r. © Bibliothèque nationale de France. 2022.

Analyzing these two kinds of dome images in Islamic manuscripts leads us to two conclusions. First, both the imagery of domes relate to Islamic cosmology, and the sightseeing of domes were probably (and may still be) triggering cosmological representations for Muslim viewers. Second, domes became the standardized form of monumental architecture in pre-modern Islam, as shown by pilgrimage illustrations. Among the illustrations of religious scenarios in *Futūḥ al-Ḥaramayn*, the most important pilgrimage destinations are shown in a top-view way, remarkably with Mecca and Medina, and in some manuscripts, Jerusalem. Nevertheless, we still do not know how the dome representations connect to the rituals performed under it, and in the mosques how rituals and representations are combined by practitioners. Here, we need to start from the basic fact that believers prostrate under the dome, face to Mecca, in order to worship God. In what follows, I will thus further investigate Islamic cosmology in association with the symbolism of the prostration so as to explore Dome and Ritual as an interconnected structure.

### 3. From Representation to Ritual

In his book *Cosmology and Architecture in Premodern Islam*, Akkach combined cosmology with architectural symbolism through a premodern Sufi perspective. Akkach notably focuses on the Kaʿba, and notes: "According to premodern Islamic sources, Mecca was the omphalos of the earth, and the Kaʿba was God's first house of worship. Being, so to speak, the first divine-sponsored architectural project, the Kaʿba is a key element in the interplay of cosmology and architecture." (Akkach 2005, p. 179) He grounds his analysis of the order and symbolism of the Kaʿba into textual analysis. Following Akkach's perspective, I will now mobilize some visual material to illustrate the fact that the symbolic/cosmological representation of the Kaʿba associates with the ritual circumambulation around the Kaʿba. This discussion will allow us to assert the fact that cosmological representations, on the one hand, and rituals, on the other, share common structures. Making use of the approach to "ritual-architectural event" propounded by Jones, I will analyze how the presence of the

cube and the circle within cosmological patterns have induced Ottoman worshippers to look at the Süleymaniye mosque. An example drawn from the realm of contemporary art will then suggest that patterns of cube and circle continue to shape the vision of contemporary onlookers.

### 3.1. The Circle as the Cosmos

Islamic cosmology is primarily based on the Quranic verses. For instance, the imagery of the seven heavens is recorded in Qurʾan, "It is He who created for you all of that which is on the earth. Then He directed Himself to the heaven, [His being above all creation], and made them seven heavens, and He is Knowing of all things" (Qurʾan 2:29, also see Qurʾan 65:12).[8] The absoluteness of the One God and of His creative power makes Islamic cosmology rely upon the formula of Unity in orthodoxy (Rahman 1967). As Nasr writes, "the cosmological sciences are closely related to the Revelation" (Nasr 1978, p. 1). The understanding of human and social facts, which in turn influences religious commemoration, is similarly determined by Revelation.

As we know well, the Lunar Hijri calendar emerged for commemorating Hijrah the Prophet Muhammad's migration (622 A.D.) from Mecca to Yathrib (Medina). And the month of Ramaḍān commemorates the revelation of the Holy Qurʾan from the Angel Jabriel [Gabriel] to the Prophet Muhammad. The Muslim calendar combines the commemoration of religious events and lunar cycles. Catherine Bell notes: "just as rites of passage give order and definition to the biocultural life cycle, calendrical rites give socially meaningful definitions to the passage of time, creating an ever-renewing cycle of days, months, and years" (Bell 2009, p. 102). These calendrical rituals are profoundly inserted into everyday life, cover all the aspects of existence, and interlace with extensive astronomical and geographical knowledge. The Islamic calendar, based on a knowledge of astronomy, not only determines the dates of Ramaḍān and other festivals through the observation of the moon, but also, similarly, the measurement of the hours of day and night determines the time of the five daily prayers. Additionally, geographical knowledge helps one to accurately determine the direction of Mecca, orienting rituals such as worship, animal slaughter, and burial.

The development of Islamic science benefited from the quest for precision in ritual acts; it is well known that the earliest astronomical observatories appeared in Islamic science (Black 2016, pp. 26–28). To determine the orientation of Kaʿba, Muslims invented the qibla compass, endowed with capability for precise calculations. David King, a scholar of the history of Islamic science, points out this double origin: "folk science" derived from the astronomical knowledge gathered by the Arabs before Islam; "mathematical science" was deriving mainly from Greek sources. It was involving both theory and computation. The former was advocated by legal scholars and widely practiced over the centuries. A select few practiced the latter (King and Lorch 1992, p. 189).

In astronomical and geographical manuscripts, celestial bodies appear in the form of circles. In the 13th century, Zakarīyā ibn Muḥammad ibn Maḥmūd al-Qazwīnī (d. 1283 A.D.) composed a book entitled *"The Wonders of Creation and Strange Creatures"* (ʿAjāʾib al-makhlūqāt wa-gharāʾib al-mawjūdāt), an encyclopedia that is one of the most remarkable books of Islamic cosmography. It has been translated from Arabic into Persian and Turkish, and transcribed and preserved in various libraries and museums.[9] *The Wonders of Creation and Strange Creatures* incorporates a wide variety of astronomical, geographical, geological, mineral, botanical, animal, and ethnological content. The first part of the manuscript includes the heavenly bodies, angels, and time. In the illustrations, the angel Rukh is holding the celestial spheres (Figure 5a,b). In the Islamic celestial theory, the representation of celestial bodies as being circular is based on the Ptolemaic model or on later theories developed by scholars such as Ibn al-Haytham (965–1040 A.D.), an Arab mathematician, astronomer, and physicist (Langermann 2007). In addition to the illustration of an angel holding a celestial body, ʿAjāʾib al-makhlūqāt collected in Cambridge University Library (MS Nn.3.74) has multiple images of the celestial spheres in circles.

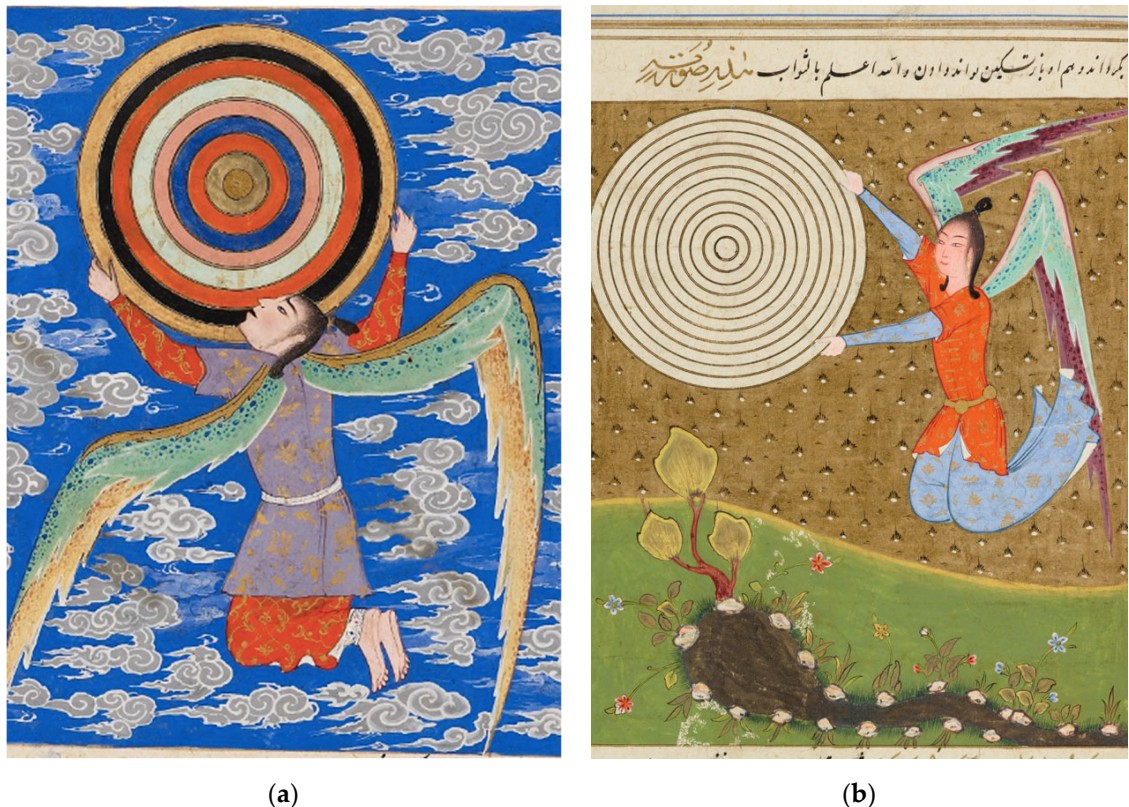

(**a**)　　　　　　　　　　　　　(**b**)

**Figure 5.** The angel Rukh holding the celestial spheres, The Wonders of Creation and Strange Creatures; (**a**) Bodleian Library, University of Oxford, EA1978.2573. © Ashmolean Museum, University of Oxford. 2022 (**b**) ʿ*Ajāʾib al-makhlūqāt*, 1566, Cambridge University Library, MS Nn.3.74, 33r. © Cambridge University Library. 2022.

### *3.2. Kaʿba, Geography and Prayer*

Another constitutive element of Islamic cosmology is sacred geography, with the Kaʿba, the black square, located at its center. This type of sacred geography appeared in the 10th century, and it remained popular until the Ottoman period in the 17th century, alongside geographical science based on precise calculations. David King researched these manuscripts in detail (King 2020, pp. 91–141; King and Lorch 1992, p. 189). King has astutely noticed that architectural details were used to define cosmological subdivisions. (Akkach also noticed it, see Akkach 2005, p. 188). Thus, while the four walls and four corners of a building indicate a division of the world into four or eight sectors, giving rise to a number of four- and eight-sector schemes, features such as the waterspout on the northwestern wall and the door on the northeastern wall were used to demarcate smaller sectors. In this way, the sacred geography of the inhabited parts of the earth comprised a variable number of sectors (jihah or hadd), all directly related to the Kaʿba (King and Lorch 1992, p. 190) (Figure 6a,b). We can find a number of these sacred geography diagrams constituted by circles around the cube in the center (Porter 2012, p. 65; King 2020, p. 165).

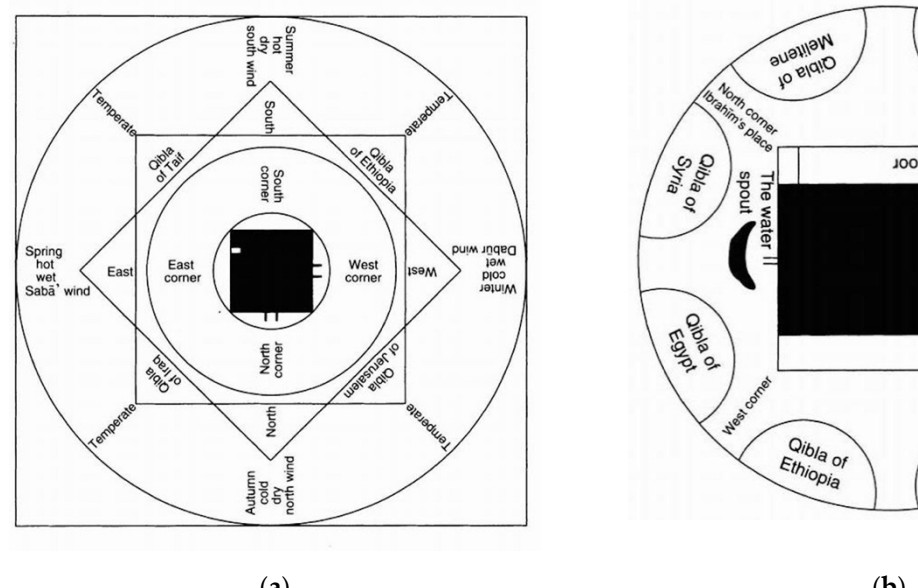

(**a**)  (**b**)

**Figure 6.** (**a**) Four-sector schemes of the Islamic sacred geography (King and Lorch 1992, p. 191); (**b**) eight-sector schemes of the Islamic sacred geography (King and Lorch 1992, p. 193). 2022.

Maps used for determining the direction to be observed while praying directly to the Kaʿba, as do the qibla walls in all mosques. And Muslims who worship under the dome in front of the mihrab undoubtedly head towards Mecca. Qurʾan (2:144): "We have certainly seen the turning of your face, [O Muḥammad], toward the heaven, and We will surely turn you to a qiblah with which you will be pleased. So, turn your face [i.e., yourself] toward al-Masjid al-Ḥarām. And wherever you [believers] are, turn your faces [i.e., yourselves] toward it [in prayer]." Besides orienting the prayer, the qibla is also used for directing burial and for implementing a taboo related to urinating. Simon O'Meara, in his book *The Kaʿba Orientation*, widely discussed material culture related to the Kaʿba, and marked the Kaʿba a as "tectonic zero" of the visuality in Islam; "Kaʿba is to Islam what the vanishing point of linear perspective is to modernity" (O'Meara 2020, pp. 123–24). These sacred maps, with the Kaʿba as a square in the center and the cosmos represented through circles around it, inevitably remind us of the illustrations of al-Masjid al-Haram in *Futūḥ al-Ḥaramayn*, discussed above: Kaʿba as the locus of orientation with all the buildings in the picture pointing towards it.

*3.3. The Cube and the Circle as Structure of Pilgrimage*

*Futūḥ al-Ḥaramayn* had a large number of transcriptions in the 16th–19th centuries, especially after the Ottoman Empire monitored a settled pilgrimage route to Mecca (Figure 7b). The diagram refers to al-Masjid al-Haram, which not only appeared in manuscripts but also in handscrolls of souvenirs and tiles. Several mihrabs in the Topkapi Palace's Haram also contain this diagram (Figure 7a).

As noted above, two enclosures surround the al-Masjid al-Haram, and the names are signed in Persian or Arabic next to the arched gates. The hanging lights of the corridor direct towards the Kaʿba. All the small buildings inside the enclosure point to the black square. The outer part of the Kaʿba is distinctly drawn in the form of a circle, referring to maṭawāf, the place to practice circumambulation. The scheme shows the Kaʿba as the core and emphasizes the interior and exterior division, the interior being loaded with the utmost level of sacredness. The stress on such structural elements becomes even more potent in pilgrimage materials, generally more straightforward, sometimes rough, in their techniques of representation. One commemorative handscroll offers a most remarkable sample: The circular area of the maṭawāf is highlighted, and the passageways, painted in red color, lead from the enclosure to the maṭawāf and the Kaʿba (Figure 7a,b). Noticeably,

the craftsman has added many red dots, such enhancing the orientation of the maṭawāf area in the image (Figure 7b).

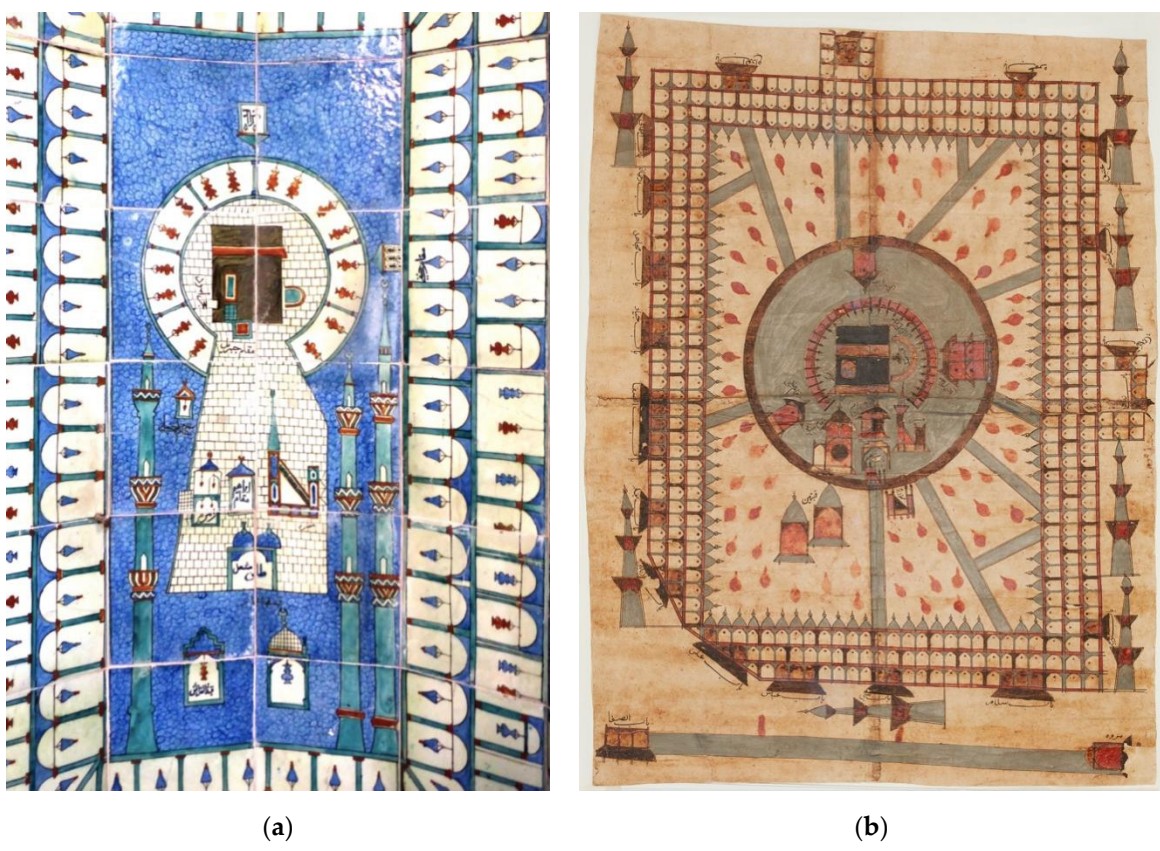

(**a**)　　　　　　　　　　　　　　　　　　　　　　　(**b**)

**Figure 7.** (**a**) Mihrab in Topkapi Palace, photo of the author, Istanbul, 2020; (**b**) Memorial scroll of pilgrimage, Saudi Arabia, probably Mecca/Hijjaz, late 18th century paper; watercolor and ink, 61.5 × 85.0 cm, Aga Khan Museum, AKM529 © The Aga Khan Museum. 2022.

　　　Another pilgrimage book *Dalʾil al-khayrāt* was composed by a Moroccan Sufi scholar, Muhammad al-Jazūlī (d. 1465 A.D.). Using a familiar model, a double-page illustration, this prayer book offers a representation of Medina and Mecca. The Aga Khan Museum collection has three manuscripts of *Dalʾil al-khayrāt*, with illustrations in diverse styles. One of them has a bird's-eye view composition to show the Mosque of the Prophet in Medina and al-Majid al-Haram in Mecca, and depicts the dome of the Mosque of the Prophet in blue (AKM382) (Roxburgh 2011, pp. 40–47; see also Porter 2012, p. 54). Another made in North Africa looks more like charts than full-fledged images, such as the ones found in the illustrations of *Futūḥ al-Ḥaramayn* (O'Meara 2020, pp. 122, 126–27). This double-page image of *Dalʾil al-khayrāt* applies contrasting colors of green and red and an intensive use of gold. The depiction of Medina includes a staircase, which refers to the minbar in the mosque of the Prophet; on the left are three staggered rectangles framed within an arched window-like scheme. Al-Masjid al-Haram in Mecca is depicted on the right page. The rectangular Kaʿba occupies the center; the four maqams or pavilions representing the four legal schools of Islam are drawn as crescent moons.[10] If we compare this four-sector schema to the 18th-century sacred geography diagrams (Figure 8), we find that they are similar in the way general orientation, the circles, and the four-sector area are underlined. Even though the illustrator may have not referred to geographical manuscripts, the symbolical space constructed by different texts (be they geographical or pilgrimage-related in scope) happens to be similar.

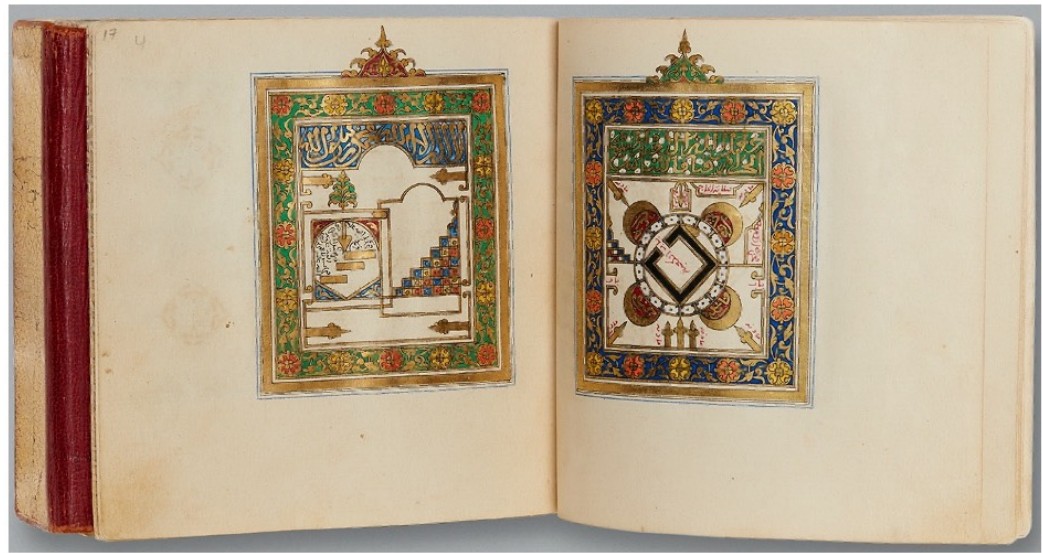

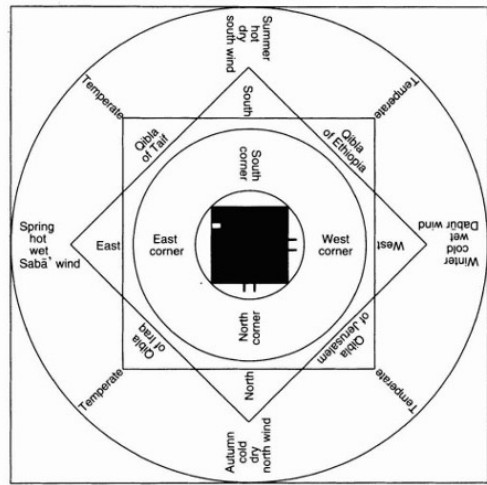
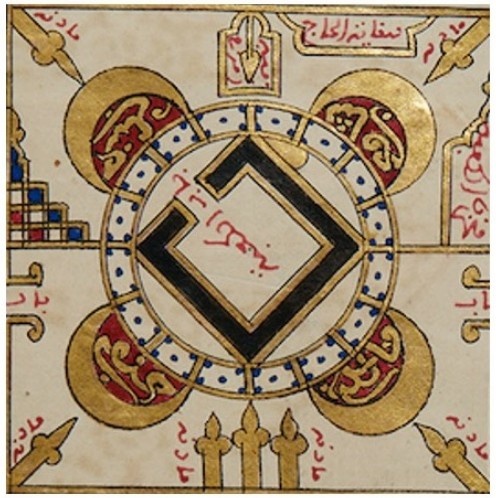

**Figure 8.** *Dalʾil al-khayrāt*, 19th century (**above**). Manuscript of Dala'il al-Khayrat Prayerbook, North Africa, probably Morocco, 19th century, opaque watercolor, gold and ink on paper, H. 13.2 cm × W. 13.8 cm × D. 5.9 cm, AKM535 © Aga Khan Museum. Licensed under CC BY-NC 2.5 CA, Comparison between the image of al-Masjid al-Haram in *Dalʾil al-khayrāt* and the four-sector Islamic sacred geography (**below**).

The Ottoman world provides us with examples of the way viewers/participants experience architectural forms from the background of preexisting cultural patterns that awaken in them a specific flux of meaning: the Ottoman scholars Nişancı Mehmed, Mustafa ibn Celal, and Sai Mustafa Çelebi compared the Süleymaniye mosque to the Kaʿba. In one of his poems, Sai wrote:

This well-proportioned mosque became the Kaʿba

Its four columns became the Prophet's four friends,

A house of Islam supported by four pillars,

It gained strength through Prophet's four friends.

(Neçipoğlu-Kafadar 1985, p. 106; Morkoç 2009, p. 205)

Both Neçipoğlu and Morkoç use this poetry to explain the importance of the four columns in the Süleymaniye mosque, and make inferences to Justinian, who had transported marble columns from antique sites to the Hagia Sophia. Morkoç applies Jones' theory to the analysis of Ottoman narratives, associating architecture with human experiences embodied into different acts, rituals, opinions, and events (Morkoç 2009, p. 200). As interesting as is this approach is, it does not allow us to directly answer the question: what made the Ottoman viewers associate the mosque and the Kaʿba?

Hagia Sophia was sharing a similar architectural form with the Süleymaniye mosque. More precisely, Mimar Sinan learnt to build the mosque from his study of Hagia Sophia. In the Christian tradition, the dome suggests a cosmic tent, a heavenly vault, and the church is a replica as the universe, while the four arches were considered at that time to correspond to the four sides of the earth. Hagia Sophia, with its great dome with four arches, was providing an occasion to present a clear image of the celestial home when homilies on God's creation were given (Smith 1971, pp. 88–89). It is worth noting that, in the early 16th century, after converting Hagia Sophia into a mosque, Ottoman writers compared the special sanctity of Hagia Sophia to that of the Kaʾba and of the Aqsa Mosque, and referred to Hagia Sophia as a second Kaʾba for the poor who could not afford the pilgrimage to Mecca (Neçipoğlu 1992). The Ottoman viewers had a different experience than other Muslim believers of similar forms of building, namely a different experience of the mosque as the Kaʿba; this is because of contextual specificities.

In the poem by Sai Mustafa Çelebi that compares the Süleymaniye mosque with the Kaʿba, the comparison is based on the architectural form and on the surrounding environment, which is a fact also shown by visual materials. First, the baldaquin of the mosque, a dome with four pillars, constructed the space with circles and the square. Second, as Neçipoğlu pointed out, "Enclosing the mosque in a wide outer precinct where caravans would pitch their tents and surrounding it with four madrasas resembling those he built around the Kaʿba and dedicated to the four Sunni schools of law reinforced the analogy" (Neçipoğlu-Kafadar 1985, p. 107). This circle, square, and the four-sector setting is like the diagram of the Kaʿba in *Dalʾil al-khayrāt* (Figure 8). Therefore, I suggest that this presence of the same pattern in different visual materials goes beyond the mere reference to a "context", and evokes a specific *architectural experience*.

The imbrication of the cube within a circle remains present in contemporary imagery. Ahmed Mater made a photogravure etching titled *Magnetism* in 2011, which forcefully expresses and reinterprets the traditional iconography of ṭawāf (Figure 9). The material used for producing the photographic effect is a square magnet. Around it are iron filings attracted by the magnet. Several Arab art galleries and collections such as the Khalili foundation of Hajj Art include this artwork, seen as exemplifying contemporary Hajj art. The artwork was also shown in the exhibition "*Hajj: Journey to the Heart of Islam*", organized by the British Museum in 2012 (Porter 2012).

*Magnetism* is reminiscent of the power attached to the Kaʿba in Mecca, particularly for Muslim viewers. The artist describes his work as follows: "Iron filings radiate around a black cube, an emanation of attraction that evokes a congregation of pilgrims thronging the Kaʿba. The unseen pull of Islam's holiest site is made manifest in this moment of absolute equilibrium. The elusive draw is faith-driven, suggestive of the deeply spiritual force felt by the millions who pray in its direction five times a day, as well as those who circle during ṭawāf."[11] Several contemporary time-lapse photographs of the ṭawāf site show shadows of pilgrims moving in the way shown by the artist.[12] The simplicity in colors and form (black and white; the circles and the square without any other element) highlights the structure of Islamic cosmogony. Note that the circle symbolizes an architectural pattern whilst at the same time it simulates the circumambulation accomplished by the believers.

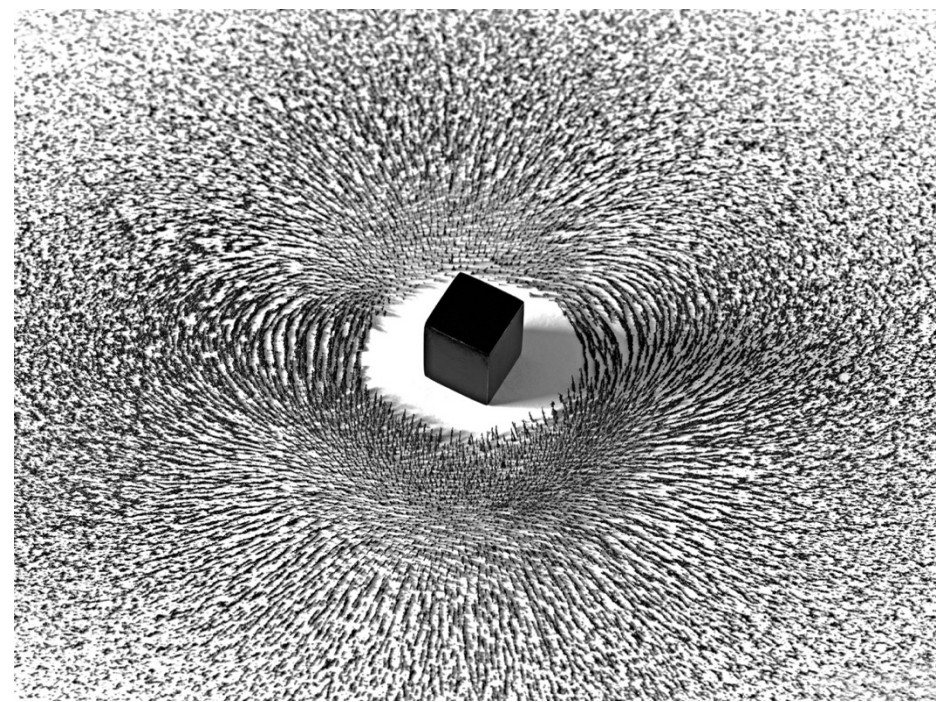

**Figure 9.** *Magnetism IV*, photogravure etching, Ahmet Mater, 2011. © Ahmet Mater 2022.

Summing up, the circle and the square constitute the dominant forms in Islamic cosmology. The celestial bodies are illustrated as concentric circles; in sacred geography, the cube is located at the center, surrounded by concentric circles. It is not a coincidence that Haft Paykar registers a cosmological correlation between architecture (the dome), astronomy (the planets), calendar (the days of the week), and geography (the princesses coming from different regions); a mental ring structure underlies the whole, and the same structure appears to determine ritual evolutions. Still, we will need to further understand and detail the way imagery, cosmology, and ritual practices are concretely correlated.

## 4. Circumambulation and the Spiritual Experience Evoked by Domes

### 4.1. The Experience of Circumambulation

The artwork Magnetism clearly hints at the fact that sacred geographical maps and ritual practices such as prayer and circumambulation obey a common schema: the Kaʿba as the cube and the cosmos as the circle. Circumambulation in Semitic tradition possesses a lengthy history: circling the "House of God' is circling the Axis Mundi (Fenton 1996, 1997). Circling the Kaʿba (in Arabic "ṭawāf") is a compulsory ritual, repeatedly performed during the Hajj, which directs the way Muslims circumambulate the Kaʿba seven times in a counter-clockwise direction. The Qurʾan (22:26) (2:125 it contains) mentions circumambulation: "Do not associate anything with Me and purify My House for those who perform ṭawāf and those who stand [in prayer] and those who bow and prostrate." Hajj and ṭawāf are so central that, in many Islamic countries, ṭawāf is broadcasted live on television. The site where circumambulation occurs is called "maṭawāf (the place of ṭawāf)".

The great Sufi master Ibn al-ʿArabī, who elaborated on ritual as mystical experience, completed *The Revelation of Mecca* (*Futūḥāt al-Makkīyah*), a vast encyclopedia of Islamic science comprising 560 chapters in the year 1260; this was shortly before he reached the end of his life (Chittick 1989, pp. x–xv). Eric Winkel published a new English translation of *The Revelation of Mecca* in 2019, including the description of ṭawāf it contains. After Ibn al-ʿArabī introduces the book's purpose as to convey to the reader "the mysteries between me and him", the opening lines of the poem read as follows:

during the ṭawāf,

'Why shall I circle,

as it is blind to perceiving our inner selves–

Petrified, an unintelligent rock with no recognition of

my circling movements?'

Then was said: 'You are totally confused, you

have lost out!

Just look at the House! His light streams

to purified hearts, bared, exposed to the light.

They see him by means of God without a veiling

curtain,

as his inner self begins to shine forth, elevat-

ed, lofty.

He shines brilliantly with tajallī to the hearts from

the horizon of a majestic

true Moon which never experiences eclipse

. . . . . . (al-ʿArabī 2019, p. 132)

In the mystical experience of circumambulating the House as described by Ibn al-ʿArabī, several images often used in Islamic mysticism emerge, such as light, veil, and moon. The light of Allah illuminates the person who circumambulates the celestial room; his heart reaches the Sublime, the veil of Allah is lifted, and the truth comes out. In another word, the circumambulation of the Kaʿba in its core is a mystical ritual directed towards the communication with the Divine.

Ibn al-'Arabī's spiritual teachings and mystical texts are often combined with embodied experience. James Morris remarks: "Chapters 66–72—one of the most fascinating and potentially valuable sections of the entire *Al-Futûhât* offer what is almost certainly the most detailed and exacting phenomenology of spiritual experience in the Islamic tradition" (al-ʿArabī 2002)[13]. Chapter 72 narrates the fact that, during the circumambulation of the Kaʿba, Ibn al-ʿArabī witnessed the miracle of the manifestation of God as a youth, the divine companion. When the circumambulation was over, the two entered the Kaʿba together (Corbin 2008, pp. 328–33; Fenton 1997, p. 363). O'Meara listed more medieval Muslims who had mystic experience from ṭawāf in his discussion on Ibn al-ʿArabī's experience of ṭawāf. (O'Meara 2020, pp. 91–94). Meanwhile, O'Meara mentioned "the ritual-architectural event" to emphasis the physicality required by ṭawāf, which is "widely regarded as bodily demanding" (O'Meara 2020, p. 96). I agree with this statement, nevertheless, reiterating the representation of Kaʿba in the pilgrimage and sacred geography manuscript, and the Ottoman's view of the Süleymaniye mosque's dome already discussed above, I suggest that there is a specific pattern shared between rituals and buildings that provide the context for these embodied experiences. A mental habit or a visual memory inscribes a common pattern into different materials. This structural feature goes beyond the specificities of a given cultural context. And it is this structural feature that enables the association between the Kaʿba and the dome.

### 4.2. Circling Back to the Dome

In the early Islamic period, circumambulation did occur in buildings other than the Kaʿba, the pilgrimage to some shrines sometimes replacing the pilgrimage to Mecca (Fenton 1997, p. 360). When Umayyad Caliph ʿAbd al-Malik (d. 705 A.D.) built the Dome of the Rock, he also built up a 'maṭawāf' around the rock, an area for circumambulation, and constructed Jerusalem as a primary place of pilgrimage (Fenton 1997, p. 361). The ritual directly requires a specific architectural form, and the architectural form symbolizes the ritual. We also see this from the illustration of al-Masjid al-Haram in *Futūḥ al-Ḥaramayn* (Figure 2a,b)

and the Kaʿba diagram in Mihrab and Memorial scroll of pilgrimage (Figure 8a,b), all of which evidently use a circle to indicate the 'maṭawāf'.

It has been suggested that the design of the dome and the spiral figuration of the mosque complex together point towards rotation, and the meaning attached to this particular movement. The Turkish historian of architecture Jale Nejdet Erzen has such written:

"The first [function] would be that sacred or religious spaces are not properly entered directly, but only after respectfully circling around them. Another more subtle meaning concerning the use of spiral forms would be how the movement of the body and its experience could refer to both cosmic and spiritual movements." (Erzen 2011, pp. 129–30.)

Erzen has used as an example the complex of the Süleymaniye mosque, which contains different public buildings, such as the mosque, the medreses (colleges), the hamam (bath), the hospital, the hospice, the soup house, the stables, the latrines, and several courtyards in between, arranged in an enveloping concentric circular form (Erzen 2011, p. 129). This concentric circular form does not appear very clearly to us. However, for the Ottoman viewers, who compared the mosque to Kaʿba, the surrounding of the mosque could be abstracted to the figure of a circle (Figure 8).

In the light of the insights brought by Islamic cosmological iconography, bodily movements around the mosque would evoke the experience of circling the Ka'ba, this particular experience deriving from the circular design of the dome. The viewing of the dome already points towards (and may awaken) the sacred experience linked to circumambulation.

We ascertained the symbolism of the Islamic dome in connection to Islamic cosmology, looking first at the poetry of Nizami. Seeking cosmological clues in the images of sacred maps and pilgrimage manuscripts, we further discovered that the combination of the circle and the square structures Islamic graphic representations. Not only is the Axis Mundi reflected in the images, but it also impacts the actions of the believers. This points towards an experience that may be awakened in the mind and the bodily senses of the viewer/practitioner: when Muslims stand under a dome, in front of the mihrab, thus facing Mecca, and when they behold the dome under which they stand, the view of this circular space possibly translates into a kind of mental and spiritual circumambulation.

## 5. Conclusions: Experiencing Sacred Architectural Places as a Structure of Meaning

We remember that Lindsay Jones has proposed to see the meaning awakened by a sacred building as a "ritual-architectural event", which incorporates building and viewer in a ritual situation. In Jones' theory, this corresponds to a total hermeneutical situation—the ritual-architectural event being constituted by (1) a human being, (2) an architectural monument, and (3) an occasion that draws the person and this monument into interaction (Jones 1993a, p. 214).

The "ritual-architectural event" can be seen as a triangular model of the interplay between people, ritual occasions, and architectural forms (Scheme 1). In this model, Jones argues that meaning arises from the interconnection between the ritual situation, the architectural space, and the viewer/participant within "a continuity of tradition".

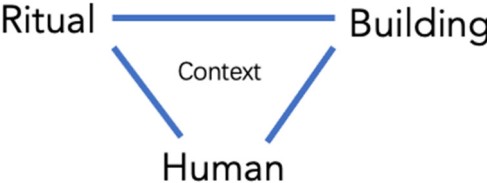

**Scheme 1.** The model of "ritual-architectural event", by the author.

Although "Humans" occupy only one corner of the "human-ritual-architecture" triangle, they are the subject of the ritual and are at the centre of the event that occurs in a given cultural context. Meanwhile, there are separate linkages between ritual and building, human and ritual, human and building:

1. The ritual determines the function and layout of the building, and the building provides the place for the practice.
2. Humans look at sacred buildings even when rituals do not take place.
3. Humans also perform some rituals outside designated buildings.

With this model, the "architectural-ritual event" generated by a domed mosque would be represented as in Scheme 2, the specific situation of this specific "ritual-architectural event" being characterized by the fact that Muslims under the dome pray towards the Ka'ba.

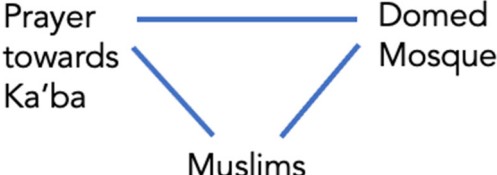

**Scheme 2.** "Ritual-architectural event" generated by a domed mosque, by the author.

However, the consideration developed in this article, the association between images, cosmology, and circumambulation, suggests a more complex structure of meaning. Scheme 3 "constructs" meaning by taking into consideration the importance of the presence of a dome in the mosque:

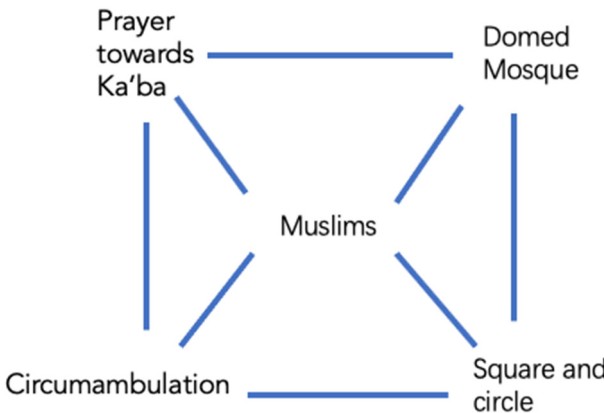

**Scheme 3.** The structure of meaning created by the presence of a dome in front of the mihrab, by the author.

What has just been said for a specific architectural form linked to the performance of a specific ritual can be generalized in the following way:

In the model (Scheme 4) here suggested, the connections between ritual and action, on the one hand, and building and mind, on the other hand, are organized into symbolic structures that support the performance of "ritual-architectural events". An implication of this model is that "ritual-architectural events" are not isolated, random occurrences, but rather are produced within an established symbolic pattern of meaning production. In other words, "ritual-architectural events" happen in the continuity of a tradition: the new experience happens within already existing patterns. Note that our approach differs from the ones that merely focus on the symbolic meaning of religious building (Kieckhefer 2014, p. 210; Smith 1971 on the symbolism of domes). While it recognizes the importance of identifying the symbolism located into the patterns, it clearly identifies the focus on the participant/viewer as key for determining the meaning taken by a building in the context of a particular event.

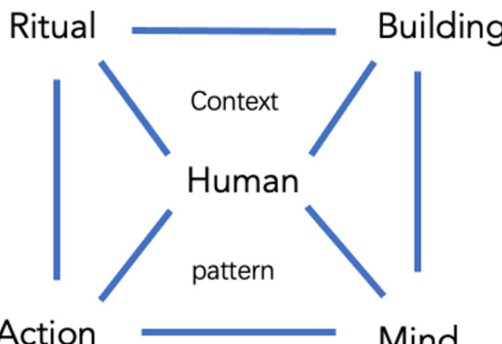

**Scheme 4.** The structure of meaning generated by the correlation between sacred architecture and ritual performance, by the author.

This model presents the participants as located at the center of the production of meaning: any one of the four triangles shaped by this model can be considered independently: human–ritual–building, human–ritual–action, human–building–mind, and human–action–mind. The consideration of these triangles will inspire my three final remarks:

1. In the structure of meaning generated by the interaction with sacred buildings, the epicenter lies in the "human–action–mind" triangle, a given cultural/religious context that determines the imaginary triggered by looking at the building and performing rituals within it.
2. The triangle "human–ritual–action" does not only refer to rituals performed in religious buildings; it also applies to ritual acts held in other settings. In this particular dimension, the stress is on a specific structure of action that determines a ritual performance.
3. Conversely, the triangle shaped by the relationship "human–building–mind" does not focus upon ritual performances but rather upon certain ways of looking at architecture.

Said otherwise, it is the taking into action of our four triangles that specifies the set of specific occurrences that this article has endeavored to study. The meaning attached to sacred architecture places is triggered by a complex of interactions between patterns referred respectively to the mind, to bodily actions, and to cultural settings. Specific "ritual-architectural events" actualize the encounter between these different patterns, but such events cannot be understood outside the patterns that preexist them.

**Funding:** This research received no external funding.

**Institutional Review Board Statement:** Not applicable.

**Informed Consent Statement:** Not applicable.

**Data Availability Statement:** Not applicable.

**Conflicts of Interest:** The author declares no conflict of interest.

## Notes

1 There are a few differences between among the several translations of Haft Paykar and researches about it, in Wilson's translation, the seven domes' representation are the following: 1st. Hindustan Assigned to Saturn; 2nd. China and Khatā. to Jupiter; 3rd. Turkistan to Mars; 4th. Irāq and Khurāsān to the sun; 5th. Transoxiana. To Venus; 6th. Rüm (the Eastern Empire) to Mercury; 7th. The hyperborean regions to the moon. (See Nizami 1924, p. 30) And in Cross' research, the second day is the dome of Rome, and the sixth day is the dome of China. (Cross 2016, p. 60).

2 For the symbolism of "7" in Persian culture, see also https://iranicaonline.org/articles/haft, (accessed on 1 April 2022).

3 On the Brethren of Purity's concept of number, see (Nasr 1978, pp. 96–97); Nasr wrote, The number 7 plays a central role in Ismāʿīli cosmology (there being seven "original" imāms and seven cycles of history), (Nasr 1978, p. 97); al-Bruni's explanation on seven climates, which has a table to show the ascensions of the signs at the equator and in the middle of each of seven climates, see (Biruni 1934, pp. 242–43).

4    The root of the word 'ilm occurs in 13 per cent of the Quranic verses, a total of 811 occurrences (El-Tobgui 2020, p. 23). The definition of 'ilm in the religious science see (Rosenthal 2007, pp. 46–70).

5    The description of the artwork *Futuh al-Haramayn* on the page of The Metropolitan Museum of Art notes: "Two details indicate that the manuscript must have been painted in the late sixteenth century: the first gate into the enclosure is labelled "blocked" (sadda), thus reflecting the closing of this gate sometime between 1569 and 1573, and there are seven minarets, thus including the one added by the Ottoman sultan Suleyman in 1565–66." See: https://www.metmuseum.org/art/collection/search/456969, (accessed on 1 April 2022).

6    In a page of *Futuh-i Haramayn* (Figure 3b), it signed Persian words different from "qubba" next to the two mausoleums, one mausoleum is in the upper left corner and another one is in the lower right corner. Moreover, the remaining domes in the diagram are named "qubba", also referring to the graves of various saints. It is noticeable that the mosque in the upper right-hand corner has no dome.

7    Christiane Gruber in her article "Signs of the Hour, Eschatological Imagery in Islamic Book Arts" studied these two illustrations. She writes: "The Dome of the Rock flanked by the scales of justice (tawāzīn) and Kawthar Pond, [ . . . ] a number of additional graphic markers and captions, such as the scales of justice (mīzān) and the bridge over hell (pul-i ṣiraṭ). Still others identify the mount as the "Rock of God" (ṣakhrat Allāh), from which it is believed that both God and Muhammad ascended into the heavens". (Gruber 2014) Rachel Milstein also published the image of the Dome of the Rock in *Kitab-e Shawqnameh*, which is shown in octagon from the top view. (Milstein 2014, p. 193).

8    All citations of the Qurʾan in this article refer to Saheeh International English Translation, see https://quran.com, (accessed on 1 April 2022).

9    A number of libraries and museums collect ʿAjāʾib al-makhlūqāt wa-gharāʾib al-mawjūdāt, for example, the British Library Or 14140, Bodleian Library, University of Oxford EA 1978, Cambridge University Library MS Nn.3.74, Aga Khan Museum AKM 367, Metropolitan Museum of Art 45.174.17. The research on the manuscripts of this book is rich: see (Berlekamp 2011, pp. 6–8, 17–18; Zadeh 2010, pp. 21–48; Carboni 1989, pp. 15–31; 2015, pp. 13–22).

10    Deniz Beyazit, Dalʾil al-khayrāt Prayer book: https://www.agakhanmuseum.org/collection/artifact/dala-il-al-khayrat-prayer-book-akm535, accessed on 1 April 2022.

11    The artist's personal website: https://www.ahmedmater.com/artworks/magnetism, accessed on 1 April 2022. The artwork's explanation is also cited in *The Art of Hajj* (Porter 2012, p. 252).

12    The time-lapse photographs or video of the ṭawāf site are available online, for example, https://www.istockphoto.com/photo/kaaba-mecca-gm601375684-103413957 (accessed on 1 April 2022).

13    James Morris wrote the Introduction of the translated and edited version of *The Mecca Revelation* in 2002 (al-ʿArabī 2002). Also see Online resource: James Morris, "Introduction to The Mecca Revelations", Online source: https://ibnarabisociety.org/introduction-to-the-meccan-revelations-james-morris/, (accessed on 1 April 2022).

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
