# Peer review of "Representing and Experiencing Islamic Domes: Images, Cosmology, and Circumambulation"

_religions, doi:10.3390/rel13060526_

Round 1
Reviewer 1 Report
The point of this article is that the domes of mosques function as a symbol of the cosmos centered around the Ka'ba and of the ritual of circumambulation that takes place around the Ka'ba in Mecca. Thus, the site and experience of a dome in a mosque may awaken spiritual experiences such as some Muslims associate with the circumambulation of the Ka'ba. This attention to the symbolism of the dome, and of the circle in the square, as developed in Islamic thought and art is presented as being in tension with the ritual-architectural event approach of Lindsay Jones. Thus the author argues that rather than just considering the triad of architecture-humans-and event, "mind" and "actions" (in this case rituals that take place elsewhere) must also be considered.
Concerns:
The discussion of the symbolism of the dome, circle, and square in circle is fine. It is clear, logical, and persuasive. It is quite plausible that Muslims may bring all this to mind consciously or unconsciously when they see or pray in a domed mosque. However, it is quite striking that the author has to leave all this at the level of possibility. The author provides no evidence showing an explicit link in Islamic thought between circumambulating the Ka'ba and the dome on another mosque. The caution displayed in the conclusion is praiseworthy, but it does limit the significance of the contribution to scholarship the author is offering.
My chief concern of the piece is that it oversimplifies Jones's model to imply that it does not allow for the kind of cultural meaning of architectural features that the author provides. A brief look at Jones's work shows that he is attentive to these cultural ideas and symbolism. To refer to the authors "Scheme 1" these are things that the human brings to the exchange between the building, the human, and the ritual. Perhaps "human" should be defined as "human and context".
The need to acknowledge that symbolic forms that reference another ritual-architectural event are important in interpreting a building is well taken. But I do not see what is gained by dividing these into "action" and "mind." I think this just gets confusing.
If Jones does not make explicit the importance of symbolic forms like domes (and I think he does), then other theorists of religious space do. See for example the discussion of "symbolism and sacrality" in Richard Kieckhefer, "Architectural Expression and Ways of Being Religious," in The Oxford Handbook of Religion and the Arts, edited by Frank Burch Brown, 204-19. New York: Oxford University Press, 2014. DOI:10.1093/oxfordhb/9780195176674.013.011 based on Kieckhefer's longer discussion in Theology in Stone.
The effort to connect this contribution to a general approach to religious architecture is to be lauded, as is the goal of improving that approach. But the theorist (Jones) needs to be treated with more care and other theories should be acknowledged. It is basically this framing of the essay that needs revision.
Other notes.
The art work Magnetism IV is a good expression the importance of the Ka'ba and the way it structures the Islamic world since every mosque is oriented toward it. It helps show that the Ka'ba plays an important role in in Islamic thought and practice even when it is not seen. I don't see how it reflects the focus on circumambulation, however, because the lines formed by the magnet are not concentric. It seems to reflect the centrality of the Ka'ba, but not circumambulation.
Several sources cited are not included in the reference list. For example, Wescoat and Ousterhout, Ragavan, Jenkins-Madina.
I am not sure what the author means by "structural" (line 55, 466, etc.). I understand how it different from the mere situational approach he ascribes to Jones, but is structural synonymous with cultural?
The point that ritual expresses important religious and cultural structures is valid, but it isn't clear how the quote from Catherine Bell supports this point 49. She is talking about how somethings are designated as ritual to distinguish them from others.
Editorial comments:
p. 2, l 46 the first word should be "do" not "to"
l. 47 don't capitalize ritual
p. 5 line 189 "Met" should be replaced with the fullname of the museum which follows in parentheses.
Author Response
Thank you very much for your reviews of my paper. Your comments helped me very much to improve my writing. I marked the changes in the reuploaded paper in red.
I received three reviews, and I summarize their comments and my answers in the file. Please see the attachment.

Reviewer 2 Report
I was pleased to see mention of the work of Lindsay Jones at the beginning of this paper as his work is not drawn on and perhaps not familiar to many who are looking at Islamic art history or architecture. If his work were introduced this might bring a fresh perspective that might be helpful in opening out new areas of research and injecting life into well-rehearsed arguments in the field. However, it became clear on reading further that the author was not interested in drawing on, developing, or applying Jones’ methodology to the Islamic dome (the subject of the paper) and eventually I began to realise that the author was keen to revise Jones’ work altogether. While Jones (drawing on Gadamer) seeks to suggest that the hermeneut/viewer/user of the work of art (or architectural space) in experiencing the work partakes in a dialogue which includes ritual, user, and building, the author, in the conclusion to his introduction suggested interest in uncovering ‘structures of meaning’ which I assume the author interprets as pre-existing and perhaps universal. What is meant by ‘structures of meaning’ is not defined and further exploration or definition or exemplification would be helpful. I also feel that a more thorough presentation and targeted critique of Jones’ theoretical development of Gadamer’s work in an introductory section would help the reader to better understand the intentions of the author.
Below I will go through to offer some ideas of where this paper might be improved. However, I feel that perhaps this paper represents the beginning thoughts around what may be a long-term piece of work for the author (I will comment on this further in my conclusion) so I have not bothered to point out typos or offer suggestions on detailed editing, e.g., places where a revision or change in word order would improve the experience for the reader. I feel that this type of editing should come at a later stage and perhaps with a revised paper.
I begin my review by stating that I have no doubt that, as the author claims, when Muslims stand under a dome, in front of a mihrab, there is the potential for this experience to ‘translate into a kind of mental, spiritual circumambulation’. I do feel that the role of architecture and its potential to awaken such experiences is clearly articulated in Jones’ work drawing on Gadamer, I was less certain of how the author’s theoretical stance supported such experience. Thus, I feel that the paper requires major revision to best present the author’s views.
Introduction
Overall, I feel that as suggested above the introduction would be improved by a clearer statement of the author’s intentions from the outset which could be framed in relation to a critique of Jones/Gadamer. Where for the author does the methodological stance in relation to the hermeneutics of sacred architecture of Jones fall short? What is lacking in Jones’ model that the author wishes to address? How does he intend to fill this gap? This would set up what (I think) is the intention of the paper, to step away from Jones and instead suggest that there are shared structures of meaning which inform experiences of Islamic architecture as opposed to architecture/the work of art as ontological participant in a dialogue which has unlimited potentialities not confined geographically or chronologically or by other structures.
In this section the author asks which elements (of architecture?) believers construct meaning from and whether they associate meaning from rituals and from forms?
Are these questions that this paper will answer? Initially I wondered if the aim was to deconstruct the hermeneutical conversation and wished to know more about methods here? It was not yet clear how the author’s approach related to Jones’ work. An initial section as suggested above would help with this. Or if the author intends to answer these questions, methods could be described here.
Again, I was confused by the next section as the author seemed to be arguing that believers are anchored into a cultural system through which they receive perceptions, messages and imagery but this did not accord with Jones’ work. If it was made clear from the outset that the author wished to depart from Jones’ work and where this departure was leading and why, this confusion might be alleviated.
How is the word ‘structural’ being used? This word has a long history and associated debates. Where does the author fall in relation to all of this and how will this word be used here?
Again, I was confused when the author stated an intention to examine ‘the correlation that links architectural forms, ritual performance, and participants’ experience into a whole’, which also seemed to contradict Jones’ work which focuses on the ‘surplus of meanings’ the polyvalency of religious buildings and the capacity of architecture to move beyond the intentions of creators to signify in ways not previously conceived.
There is no suggestion of geographical or chronological framing here. Is the author suggesting that there is one Islamic ‘structure’ that all Muslims from the hijra onwards conform to? Also, what can be said about the fact that domes and niches with domes are extensively used prior to Islam and are widely seen in other cultures simultaneously?
Sentences like ‘the Dome [why is the D in upper case here] is loaded with significance in Islamic societies’ require discussion. What societies? Where? When? What specifically is this ‘significance ‘ that is spoken of?
The argument for the use of manuscripts ‘to direct the inquiry’ is simply that domes appear in them. Here Jones’ concept of the ‘ritual architectural event’ is used as support for this decision and defined as ‘the meaning of the building depends upon the participants, and it takes shape in the way they experience the building’. I suggest that this characterisation of ‘ritual architectural event’ does not take into account the basis of the argument which suggests that what any piece of architecture is able to ‘say’ reaches beyond any historical confinement . It also ignores Gadamer’s suggestion of the “ontological possibility”’ of architecture to ignite or participate in life transforming experiences. Jones calls for a movement away from form to consider buildings as “participants” in events. Again here further engagement with Jones’ ideas would allow the author to show their intention to depart from this way of conceiving architecture and to lay the groundwork for a contrasting approach. Jones’ two part study on ‘The Hermeneutics of Sacred Architecture: A Reassessment of the Similitude between Tula, Hidalgo and Chichen Itza Yucatan’ published in History of Religions is readily accessible and a good introduction to Jones’ work and Gadamer’s. Nicolas Davey also writes well on Gadamer’s aesthetics, which might be helpful here. At this point, as a reader, I was still unsure whether the author hoped to exemplify Jones’ work and so I felt that it would not be possible to make a connection between the manuscripts the buildings and the participants unless the author was speaking of their own experience or that of someone they observed or speculating on a potential experience. Was it the intention of the author to suggest a hermeneutical dialogue which would be deconstructed based on the author’s experience to see what resulted? It was unclear what the domes and manuscripts had to do with Jones’ methodology which encourages a move beyond consideration of particular forms. Again, it would have been helpful to know that the paper was intended as a complete revision and to hear about methodology and methods in this section.
The conclusion of this section states that the author ‘will associate these various elements, suggesting a way to depict structures of meaning associated with the fact of experiencing one’s setting into sacred architectural places.’ Apart from the construction of this sentence which made difficult to see what was being suggested, there seemed to be an unstated and palpable tension between Jones’ suggestion of a hermeneutical dialogue/play involving building, human and ritual, all as participants which leads to shifts in meaning and personal transformation. I was unclear as to what exactly the author was saying in this sentence but I began to feel that perhaps it was this tension that would be explored?
Section 2 Images of Domes in Manuscripts
2.1 Poetry Manuscripts
This section is based on the widely discussed Haft Paykar of Nizami (really worth reading Michael Barry on this poem). The author intelligently suggests that it might be possible to move beyond Grabar’s observation that the domes in the poem relate to an ‘ideology of pleasure’ and suggests analysis of the poem will show that ‘the aesthetic experience that the beholders (which beholders?) acquire from the dome goes considerably beyond pleasure’. Here ‘cosmological symbolism’ is mentioned. How will the author use this word? This requires contextualisation in the context of Islamic art and architecture. I feel it is important here to mention that it is the poet himself who draws associations between the colours of the domes (not the domes themselves), days of the week, and planets. The domes ‘house’ or create the space for the princesses.
I was unsure of the purpose of the discussion of the significance of the number seven and its association with colours, planets, days of the week? I think this discussion could either be shortened or developed but as it stands it is not clear to the reader the connections that are being made. Potentially, with a more extensive and focused discussion with relevant citations, the fact that there are seven domes could be connected with the seven heavens and therefore each dome could potentially be associated with one of these heavens, but this has not been suggested. There are endless historical sources which associate domes and canopies with heaven which could be drawn on. But again, these discussions would need to be tied in with a clearer methodology. I do not think that Jones’ concept of the ‘ritual architectural event’ would take account of historical studies into associations at particular times as he would argue that there is a hermeneutical dialogue taking place in each participant’s experience of the building regardless of context wherein a wealth of potential meanings could be drawn on and transformed in the play of association of building, human and experience/ritual.
I was not aware of Al Buruni’s comparision of the planets with architectural types, which is another example of the associative thinking of pre-Modern Islam, but it is difficult to accept the jump from Al Buruni’s association of, for example, Jupiter with royal mansions, etc. and the author’s next sentence which states that the associations between particular planets and associated colours and particular building types connect Buruni with Nizami and then ‘suggest the continuity of Islamic cosmology’. The next sentence states ‘The correlation between architectural forms (the dome), astronomy (the planets), the calendar (the days of the week), and geography (the princesses of different regions) does not accidentally appear in Nizami’s text. A consistent system of Islamic cosmology lies behind it.’ One problem with this sentence is the dome is an overarching feature rather than one of the seven differentiated things (like days, colours, forms of architecture discussed by Buruni who does not seem to mention the dome in the section on architectural form quoted here) that are associated. The dome in general is not particularly associated with one of the seven things discussed so what is the point of the discussion of the significance of the number seven? This argument breaks down because the dome is not connected with one of the particular of the seven things, it sits above all of them. Thus, it is quite difficult to understand how this suggests ‘a consistent system of Islamic cosmology’.
Then the author argues that the dome synthesizes astronomical, arithmetical, architectural, and decorative knowledge based on a very brief discussion with little citation and no specific examples, geographical or chronological references for the linguistic link between the arts and other sciences. When the author states, ‘Seen from the perspective implicit in this concept’ it is unclear what concept is being referred to. When it is stated that the imagery of the seven refers to astrology and geography, this seems clear but what about the dome itself. How is it operating in this context? There needs to be a movement between the consideration of the seven and acknowledgement that each of the seven is representing one aspect of a group of seven (planets, climes, days of week, colours). BUT the dome is the thing that is not different but instead repeated in each case. Consideration of this would be interesting. T
his section ends with the statement that the ‘aesthetic experience of the dome is not only architectural it proceeds from a synthesis of knowledge, such synthesis being provided, in pre-Modern Islam by cosmology.’ I have a few questions about this statement. First, whose aesthetic experience are we speaking of? Where? When? (Later in the section there is reference to pre-modern Islam but I believe that that is the first mention of a context for this discussion. Should this be laid out from the beginning? And if so perhaps further consideration of methodology is required.) Second, when reading this I felt that to refer to any architectural experience as an ‘only’ seems to completely contradict the terms of Jone’s ‘ritual architectural event’ which is based on Gadamer’s discussion of play and hermeneutics in Truth and Method where he emphasises that the experience cannot be separated from the form or the participant. Gadamer recognises and Jones develops the argument of the fullness and inexhaustibility of architectural experience rather than speaking of it as limited, as an ‘only’. However, this was again a clue that instead of drawing on or developing from Jones the author was suggesting a revision of thought and approach to architecture and experience. Finally, I feel that the discussion of Islamic cosmology was too brief to help a reader to come to the same conclusions. Islamic cosmology is not just the seven planets and associations. The subject requires depth.
I do not dispute that there are potential intellectual links to be drawn between arts and sciences historically in the context of Islamic thought, nor that one could argue for a pre-Modern Islamic cosmology that may have been shared over a long period of time. I am very aware that these aspects are little discussed and often ignored by many art historians discussing Islamic art. However, the argument as it is constructed here is not robust enough and I consider that it would not improve the situation. I would suggest that this argument requires perhaps a preliminary paper (or book) that then another paper could build upon.
Section 2.2 Pilgrimage Texts
Although it is historically the Temple Mount, I would refer to the platform atop Mt Moriah in Jerusalem as the Haram al Sharif in the current context.
The maps of the Haram that are referred to are not at all unusual when considered alongside most of the ‘maps’ of the period, many of which are constructed in the same way, with a birds-eye-view of cities or towns or places. I think it slightly misleading to state that ‘This unusual setting transforms the image into a diagram’ as this is a very conventional means of representation at the time, albeit diagrammatical. A brief survey of images from different geographical and religious contexts would show this. What exactly is the author trying to say here? Also, does it follow that because there are domes in these images ‘the Islamic world was landmarked by domes that had a clear monumental and sacred connotation’? Again, I do not dispute the conclusion that there were lots of domes and that they had both monumental and sacred connotations, but their appearance in a few maps is not enough to jump to this conclusion. A more nuanced argument is required.
It would be helpful to clarify the relationship of the description of the Dal ‘il al-khayrat on page 7 to either the previous discussion or the subsequent discussion. There are no connections made here.
In the first paragraph on page 8 when the author speaks of ‘a sideways form with a pointed arch shape’ would architectural terms such as ‘elevation’ and ‘plan’ be useful here? Also in this paragraph it is stated that the Dome of the Rock is represented as octagonal ‘to reflect the unique religious scenario’ however, the Dome of the Rock is an octagonal building so it is no surprise that it is represented as an octagon. Perhaps this can be clarified? The relationship between the number eight and the ‘Last Judgment’ might be of interest to the author. There is evidence for octagonal buildings referring to transitional states (to name just one example, the verses inscribed in St Ambrose’s baptistry in Milan ca. end of 5th century). A start for consideration could be the book on number by Schimmel that is drawn on earlier in the paper and consideration of Quran 40:7 and 69:17.
Section 3 From Representation to Ritual
This section would benefit from some discussion at the beginning to help the reader to understand its purpose and to show how it forwards the argument being made and/or some discussion to guide the reader to the connections between the sections and how they build on one another to make an argument that relates to and forwards the general thrust of the paper.
3.1 The Cube as the Ka’aba, the Circle as the Cosmos
Is there evidence for the statement- ‘The absoluteness of the One God and of His creative power make Islamic cosmology focus more on the observance of Ritual [why uppercase ‘R’] than on a quest of Origins [why upper case ‘O’]’ (again this could be the topic of a book and requires lengthy discussion). And is the ‘more’ used here meant to suggest comparison with something else? Perhaps the way that rites of passage, ever-renewing cycles, etc. are described Is characteristic of many pre-Modern contexts? Are there examples of the way that astronomical, geographical knowledge and calendrical rituals are related and cover all aspects of existence? It would be extremely useful to draw on supporting sources in this discussion. I would like to see more nuance. The example of Ramadan and other festivals being determined by the lunar calendar does not seem to back up the claim for profound insertion of these knowledge systems into everyday life.
I am aware I am being slightly pedantic and overcritical here but I feel that this discussion, like many aspects briefly mentioned in this paper, requires more nuance. I think there are many claims being made and wonder if they detract from the discussion in this section. For example, it is stated that the development of Islamic science benefited from the quest for precision in ritual acts. This sort of statement could be the starting point for a number of doctoral theses. Is it necessary to make this claim here without evidence or discussion? Does it really forward the argument being made? Another claim, ‘Islamic cosmology is a mixture of “folk knowledge” and scientific analysis.’ The same criticisms apply to this statement.
NB The cosmological diagrams presented follow conventions used in many other contexts where the orbits or locations of the planets in comparison to earth are suggested by concentric circles around the earth. Angels were often shown not as holding the cosmos but as setting it into motion.
3.2 Geography and Liturgy
The author may wish to reconsider the use of the word liturgy in this context or explain how/why it’s being used. Also some introduction and conclusion to this section would aid the reader.
3.3 The Cube and the Circle as Structure of Pilgrimage
I think the author might find writings by both Wm Chittick and Frithjof Schuon on the ‘Five Divine Presences’ of interest in relation to the divisions of four around a central figure. Here the author moves to images from the contemporary context. As it would be possible, it might be interesting to speak with the artist to see whether or not they recognise all of the suggested conclusions made by the author. If the author were able to sort out the methodology or to continue with the line of argument made by Jones in relation to the ‘ritual architectural event’ this would not be necessary and would mean that this could be brought in easily. However, in that case, the whole paper would require reconsideration as that line of argument is not being followed. Instead, the author seems to be suggesting that there are particular conceptions of meaning which are shared by many across geography and time. (These were my first impressions, again an introduction and clear outline of the critique of Jones’ work and the proposed contrasting approach of the author would answer these questions. However the author’s approach would need to consider in what context the contemporary work is brought into the argument as to make an argument for ‘structures of meaning’ I feel would require chronological and geographical contextualisation.)
- Circumambulation and the Imagination Evoked by Domes
(Is imagination the correct term here?)
4.1 Circumambulation as Mystical Experience
This title suggests another very large topic which to give it its due would require a clear methodological stance with related methods, careful definition of terms, review of previous work in the area, numerous examples. This section could simply be titled ‘The Experience of Circumambulation’ and then Ibn ‘Arabi’s description moves the discussion onwards as it speaks of an experience that could be described as mystical.
I would say that Ibn ‘Arabi’s texts are not ‘combined with embodied experience’ but based on his experience.
At the end of this section the author states, ‘I suggest that there is a set of structure shared between rituals and buildings that provide the context for these embodied experiences. And that it is this set of structure that enables the association between the Ka’aba and the dome.’ This seems to be the central argument of the paper but the reader at this point is not certain what is meant by ‘a set of structure’ (does the author mean ‘a set [group?] of structures ‘ or perhaps ‘a set structure’ as a set of structure is not grammatically correct.) And is it possible to argue at the same time for Jones‘ ‘ritual architectural event’ and a set of structure[s]? I think this is the major flaw at the heart of the thinking. Again, explaining that the author will not be adopting Jones’ approach from the beginning would alleviate this. This important sentence needs to be considered carefully as the construction suggests that it is the rituals and buildings that share the structures of meaning rather than the viewers or users, which I think is what the author wishes to argue.
4.2 Circling back to the Dome
It is not clear how Abd al Malik’s construction project on the Haram al Sharif in the 7th century suggests that the ritual practice of circumambulation in Islam ‘directly requires a specific architectural form’ as the Dome of the Rock is octagonal compared with the cube of the Ka’aba and is it remarkable that a circle is used to represent the matawaf when the ritual is based on a circular motion around a central point? I am being pedantic here (again) but in a paper when words are precious the all words need to be precise and well-considered.
Then the subject changes to the dome. How does this relate to the previous discussion of an octagon and a cube? And is it possible to give an example of a ‘spiral configuration’ of a mosque complex? Normally they are quadrangular and/or based on fitting into the surrounding urban environment. I have not observed a spiral configuration of a mosque complex anywhere.
This sentence is problematic: ‘In light of the insights brought by Islamic cosmological iconography, bodily movements around the mosque would evoke the experience of circling the Ka’aba, this particular experience deriving from the circular design of the dome.’
1- Why make the claim about insights brought by cosmological iconography? Is knowledge of cosmological diagrams necessary for the Muslim to recall the practice of circumambulation around the Ka’aba?
2- The use of the phrase ‘bodily movements around the mosque’ could refer to any movement of any body walking in or out of, sitting in, or doing anything in any mosque. Is this meant to refer to a specific type of movement (e.g., circumambulation)? By this is the author trying to argue that movement in any mosque can be reminiscent of circumambulation around the Ka'aba? I think that is what is being said but is not at all clear from this sentence.
3- The way the sentence is composed it suggests that the author believes that the experience of circumambulating around the Ka’aba derives from the circular design of domes. I am certain this is not what is meant but it is not clear what is meant. Perhaps that the circular design of domes reminds the Muslim of circumambulation? Or that this form (dome and drum and arcade) creates a space for circumambulation? Not sure.
Then in the next paragraph the reader is asked to agree that, based on a brief look at the use of domes in the Haft Paykar and in maps/diagrams of sacred spaces (where there actually are domes so their use is not surprising) and on consideration of diagrams of circles and squares (forms which are used extensively in diagrams from many different religions and also in secular diagrams) which are not directly related to domes (although certainly a relationship could be carefully and systematically argued for a relationship between the dome and the circle or sphere and related to the square or cube), an ‘experience may be awakened in the mind and the bodily senses of the viewer/practitioner’. But certainly, that follows directly from what Lindsay Jones has argued from the beginning, i.e. that there is a hermeneutic process that potentially arises in our experience of architectural space. So the reader wonders what the intervening study has added to the story? Here again a clear statement from the outset which suggests that Jones’ work is insufficient to encompass what the author is trying to argue for would mean that at least the last question would not arise. In terms of the other problematic areas, I feel that the author requires more discussion and more examples of each of the aspects to make a coherent argument. Perhaps each of these areas requires discussion in their own paper (book) rather than trying to put too much into a single paper.
I would suggest that the brief mention of the Axis Mundi (explain concept and relate to Islam) as reflected in images (Which? Where?) and impacting the actions of the believers (Where? When? How?) in one sentence requires further consideration and careful presentation.
5 Conclusion
In relation to the model presented by the author (scheme 4) how are ‘ritual’ and ‘action’ differentiated? Presumably the word ‘ritual’ encompasses the action taking place to carry out that ritual? This needs clarification.
The conclusion helps to clarify the intentions of the author. It is clear that he disagrees with Jones’ (and ultimately Gadamer’s) notion that the work of art is not limited to interpretation by historical context (pre-existing patterns) and instead functions ontologically in the hermeneutic dialogue and experience of the viewer/user to ‘occupy a timeless present’.
Final thoughts on this paper-
As discussed above a clearer statement of intentions, methodology, methods from the beginning and a critique of Jones’ work which states clearly how the author’s views differ is essential to this paper. However, I feel that the author is just beginning to develop the ideas and arguments that may end up being a life’s work. I would suggest that, like Jones, he start with small chunks, aspects which could be developed, small and single examples that begin to help to develop the idea of a set of structure(s) or a set structure. Doing things in this way would help the author to clarify their own stance in contrast with Jones. Or perhaps it is not even necessary to mention Jones, who is only really used to suggest that the experience of architecture and associated rituals are important to consider when we think about meaning and architeture. This argument could possibly be made using someone who also supports the author’s view that there are set structures of meaning that come into play. I cannot help but feel that the idea of ‘a set of structure’ that the author speaks about is the key aspect of their thinking but this is not clearly set out, or clarified or developed. If the author were able to take small steps to begin to develop this idea and to find texts that support this, I suggest it would be possible to base a paper, maybe using the experience of the dome as an example, or maybe circumambulation, or maybe cosmological diagrammes. At the moment the breadth of the paper is not helpful. A paper which just began the critique of Jones’ work and spoke of the idea of the set of structure(s) would maybe be a good start.
I feel that the author is not satisfied with current debate and scholarship in relation to the experience of architecture and its meaning. I agree that this is an area that requires much more work and a very interesting area to be working in. I think if the author had time to consider and develop their ideas and perhaps started with less ambitious and more succinct presentation of their ideas they could make an important and original contribution to this area.
Author Response
Thank you very much for your reviews of my paper. Your comments helped me very much to improve my writing. I marked the changes in red.
I received three reviews, and I summarize their comments and my answers in the file. Please see the attachment.

Reviewer 3 Report
Overall, this article was a joy to read. The author does a wonderful job of connecting the cosmology, mathematics, aesthetics, rituals, and emotions that come together to give domes their importance in Islamic worship. The author makes good use of theory, but makes these theories understandable even for readers who are not specialists in their immediate field.
Author Response

(The authors gave the same response as above.)

Round 2
Reviewer 1 Report
Good expansion of theoretical discussion.
copy editing note: in line 111 the second "ottoman" needs to be capitalized.